

# Wind-Driven Emissions of Coarse Mode Particles in an Urban Environment

Markus D. Petters[1], Tyas Pujiastuti[1], Ajmal Rasheeda Satheesh[1], Sabin Kasparoglu[1], Bethany Sutherland[1], and Nicholas Meskhidze[1]

[1]Department of Marine, Earth, and Atmospheric Sciences, North Carolina State University, Raleigh, NC 27695, USA.

*Correspondence to*: Markus D. Petters (mdpetter@ncsu.edu)

**Abstract.** Quantifying surface-atmosphere exchange rates of particles is important for understanding the role of suspended particulate matter on radiative transfer, clouds, precipitation, and climate change. Emissions of coarse mode particles with a diameter greater than 0.5 µm provide giant cloud condensation nuclei and ice nuclei. These emissions are critical for understanding the evolution of cloud microphysical properties yet remain poorly understood. Here we introduce a new method that uses lidar retrievals of the elastic backscatter and Doppler velocity to obtain surface number emissions of particles with a diameter greater than 0.53 µm. The technique is applied to study particle number fluxes over a two-month period during the TRACER campaign at an urban site near Houston, TX, USA. We found that all the observed fluxes were positive (upwards) indicating particle emission from the surface. The fluxes followed a diurnal pattern and peaked near noon local time. Flux intensity varied through the two months with multi-day periods of strong fluxes and multi-day periods of weak fluxes. Emission particle number fluxes peaked near ~100 cm$^{-2}$ s$^{-1}$. The daily averaged emission fluxes correlated with friction velocity and were anticorrelated to surface relative humidity. The emission flux can be parameterized as $F = 3000u^{*4}$ where u* is the friction velocity in m s$^{-1}$ and the emission flux $F$ is in cm$^{-2}$ s$^{-1}$. The u* dependence is consistent with emission from wind-driven erosion. Estimated values for the mass flux are in the lower range of literature values from non-urban sites. These results demonstrate that urban environments may play an important role in supplying coarse mode particles to the boundary layer. We anticipate that quantification of these emissions will help constrain aerosol-cloud interaction models that use prognostic aerosol schemes.

## 1 Introduction

Atmospheric particulate matter plays an important role in modulating atmospheric processes and causes changes in direct radiative forcing, warm and cold cloud microphysical structure, and precipitation initiation (Andreae and Rosenfeld, 2008; Levin and Cotton, 2009). Atmospheric particulate matter spans sizes between ~3 nm for newly formed particles and up to 10s of micron for large dust particles and bioaerosol such as pollen. Particles with a diameter > 1 µm are usually referred to as



coarse mode particles. Coarse mode particles are predominantly produced by mechanical processes such as wind-blown dust, sea spray, and bioaerosols (Horvath et al., 1990; Seinfeld and Pandis, 2016; Andreae and Rosenfeld, 2008) where emissions are controlled by wind speeds. Examples are the production of dust from eroding soils (Kok et al., 2012) and the production of aerosol from agitated ocean surfaces (Vignati et al., 2010). Coarse mode particles play an important role in the atmosphere

by providing giant (> 1 µm, Yin et al., 2000) and ultra giant (> 10 µm, Johnson, 1982) cloud condensation nuclei, which in turn may influence warm rain initiation (Feingold et al., 1999; Yin et al., 2000; Cheng et al., 2009). Furthermore, greater than 0.5 µm diameter particle concentrations have been shown to correlate with ice nucleating particle concentrations (Georgii, 1959; DeMott et al., 2010), which in turn influences first ice initiation in mixing phase clouds. Knowledge of concentration and emission fluxes is critical for understanding aerosol-cloud-climate interactions on a global scale.

Characterization of the atmospheric coarse mode particle concentration is challenging. Low number concentrations, often far less than 10 cm$^{-3}$ (Hussein et al., 2018; Moran-Zuloaga et al., 2018; Perring et al., 2015), necessitate large flow rates in particle counters to obtain good counting statistics. Aerosol inlets in airborne platforms have 50% cut sizes in the 1-10 µm diameter range and thus can artificially truncate the sampling of the coarse mode (Blomquist et al., 2001). New ground-based inlets may extend this sampling range. However, quantitative characterization of transmission efficiency remains difficult due

to instrumental constraints (Bullard et al., 2017).

Characterization of the emission and deposition rates for supermicron particles is also challenging. The eddy-covariance technique is one method to study turbulent particle transport across a dividing plane. This technique uses the covariance of vertical motion with particle number or mass and uses this quantity to derive emission fluxes. A few studies used the eddy-covariance technique to measure the sea spray aerosol flux from the bubble bursting (Nilsson et al., 2001; Norris

et al., 2012). The eddy-covariance has also been used to study deposition velocities for supermicron particles (Gallagher et al., 1997). However, most measurements in that regime have been performed using other techniques (Farmer et al., 2021, and references therein).

Light Detection And Ranging (LIDAR) is a remote sensing method that uses light in the form of a pulsed laser that can be used to measure the spatial distribution of aerosol optical properties. The absence of inlets and the potential for high-

resolution spatial sampling makes this technique attractive to characterise (aerosol) fluxes. Several prior studies used either a Doppler lidar alone, or a Doppler lidar colocated with a second lidar to estimate latent heat fluxes (Lareau, 2020; Behrendt et al., 2020), aerosol backscatter flux (Pal et al., 2010), or mass flux (Engelmann et al., 2008; Wang et al., 2021). Pal et al. (2010) suggested that fluctuations in elastic backscatter correspond to fluctuations in aerosol number concentration. These authors, however, did not further explore the possibility to retrieve aerosol number fluxes from these data.

Here we use data from a Doppler lidar to obtain the backscatter flux using the eddy-covariance technique from Doppler vertical velocity and attenuated backscatter at $z = 105$ m. Building upon the prior studies, we relate backscatter to particle number concentration by calibrating the lidar retrievals against optical particle counter-measured ground-based aerosol size distribution and radiosonde-interpolated relative humidity at the lidar sample height. Based on this calibration, aerosol





number fluxes for particles with $D > 0.53$ µm are retrieved. Fluxes over a two-month period are analysed. Implications for particle emission sources and ice nucleation particle number concentrations are discussed.

## 2. Methods

### 2.1 TRACER

The main goal of the TRacking Aerosol Convection interactions ExpeRiment (TRACER) campaign was to study aerosol-cloud interactions during deep convection over the Houston area. The US Department of Energy (DOE) Atmospheric Radiation Measurement (ARM) deployed the Aerosol Mobile Facility (AMF) at the La Porte site in Houston, TX between 1 October 2021 and 30 September 2022. The AMF deployment collected a variety of in-situ meteorological and aerosol data as well as data using multiple remote sensing platforms. Figure 1 provides an overview of the measurement site. The AMF was located

at the La Porte municipal airport, in the Southeastern region of the Houston metropolitan region. An intensive observation period (IOP) took place between 1 June and September 2022.

### 2.2 Meteorology data

The Eddy Correlation Flux Measurement System (ECOR) provides measurements of the latent and sensible heat flux, as well as the friction velocity (Sullivan et al., 2021). The instrument uses Windmaster 3D sonic anemometer (Gill Instrument,

Saltmarsh Park, 67 Gosport Street, Lymington, Hampshire. SO41 9EG, United Kingdom) and infrared gas analyzer (LI 7500 and LI 7700) to measure high-frequency correlations between vertical velocity, air temperature, and water vapour density resulting in vertical fluxes data at 30 min time resolution. The ECOR sensor height is ~ 3 m. Latent and sensible heat flux values from the ECOR are used to calculate the saturation ratio flux (Supporting Information). The ARM Surface Meteorology Systems provided surface wind, pressure, temperature, relative humidity, visibility, and precipitation measurements at 1 min

time resolution and at a sampling height of 8 m.

### 2.3 Optical particle spectrometer

The AMF housed an Optical Particle Counter (Grimm Model 11-D, GRIMM Aerosol Technik GmbH & Co.KG, Dorfstraße 9, Ainring, 83404, Germany). The size distribution is binned into 31 equidistant channels ranging from 0.29 µm to 31.15 µm. Data are reported at 6 seconds of time resolution. The instrument includes a temperature and relative humidity sensor inside

the optical block to monitor the thermodynamic state of the sample flow. The sample is dried to a relative humidity between 17 and 40% using a single tube Nafion dryer MD-700 by Perma Pure (1001 New Hampshire Ave, Lakewood, New Jersey 08701, United States).



### 2.4 Doppler Lidar

### 2.4.1 Instrument

The DOE ARM program maintains a network of coherent Doppler lidar instruments, which are manufactured by Halo Photonics (Brockamin, Leigh, Worcestershire United Kingdom WR6 5LA GB). The Doppler lidar transmits at a wavelength of 1.548 μm, with ∼150 ns (22.5 m) pulse width and < 100 μJ pulse energy at a rate of 15 kHz, providing time- and range-resolved measurements of attenuated backscatter and radial velocity (Newsom and Krishnamurthy, 2020; Newsom et al., 2017). When operated in vertical fixed-point mode, the system measures vertical velocity at 1Hz temporal and 30 m vertical spatial resolution. The lowest acceptable range gate is 105 m. The primary scattering mechanism is atmospheric aerosol. To date, the main application of this instrument has been the observation of turbulence within the boundary layer (Newsom et al., 2015; Williams and Qiu, 2023). When operated in hemispheric scanning mode, the instrument yields 2D wind fields as a function of height. The DOE ARM program collected the lidar data. The instruments are operated by DOE personnel, and data are distributed through publicly accessible archives (Newsom and Krishnamurthy, 2021; Shippert et al., 2022). Data in the archive has undergone a first pass of data processing. The vertical fixed point data files, which are primarily used here, contain attenuated backscatter coefficients, signal-to-noise ratios, and vertical velocities at approximately 1 s intervals.

### 2.4.2 Relationship between particle size distribution and lidar backscatter

The lidar backscatter is attenuated by the two-way transmission through the atmosphere. The true backscatter can be found via (Platt and Collins, 2015)

$$\beta(r) = \frac{\beta_{att}(r)}{1 - 2S \int_{r_1}^{r_2} \beta_{att}(r)\, \mathrm{d}r} \tag{1}$$

where $\beta(r)$ is the true backscatter coefficient, $\beta_{att}(r)$ is the measured attenuated backscatter coefficient, $S$ is the lidar ratio defined in Eq. (4), and the integration is carried out between ranges $r_1$ and $r_2$. The true backscatter coefficient and attenuated backscatter coefficient are equal in the first range gate. The lidar ratio represents the ratio of the extinction cross-section and 180° backscatter cross-section and varies between 5 and ~100 sr. Its value depends on the wavelength, aerosol refractive index, aerosol size distribution, aerosol hygroscopicity, and the presence of absorbing gases in the atmosphere. Most prior studies have focused on systems with wavelengths 355, 532, and 1064 nm (Tesche et al., 2009; Dionisi et al., 2018; Haarig et al., 2022). Consequently, there is limited information about values for $S$ for wavelength > 1 μm. However, the lidar backscatter and lidar ratio can be estimated from the aerosol size distribution, aerosol hygroscopicity, and Mie theory (Klett, 1985; Takamura and Sasano, 1987; Chemyakin et al., 2021; Zhang et al., 2022).



$$\alpha_{mie} = \int_0^\infty \frac{\pi D^2}{4} Q_{ext}(\lambda, m, D) \frac{dN}{d\ln D} d\ln D \tag{2}$$

$$\beta_{mie} = \int_0^\infty \frac{\pi D^2}{4} Q_{back}(\lambda, m, D) \frac{dN}{d\ln D} d\ln D \tag{3}$$

where $\alpha_{mie}$ is the extinction coefficient derived from Mie theory, $\beta_{mie}$ is the aerosol lidar backscatter coefficient derived from Mie theory, $D$ is the particle diameter (m), $Q_{ext}$ and $Q_{back}$ are the extinction and backscatter efficiencies determined from Mie theory, $dN/d\ln D$ is the aerosol size distribution in units of spectral density, $\lambda$ is the wavelength (of the lidar), and $m = n + ki$ is

the complex aerosol refractive index with a real ($n$) and imaginary ($k$) component. The integration is performed over the entire size distribution. The Mie solution is based on the assumption that the particles are spherical. The modelled lidar ratio is

$$S = \frac{\alpha_{mie}}{\beta_{mie}} \tag{4}$$

The effects of scattering and extinction by molecules are not considered here.

At elevated relative humidity particles may swell and take up water. The refractive index of the mixed particle can be
obtained from the volume-weighted average of the refractive indexes of the dry aerosol and water (Shettle and Fenn, 1979) and the water content estimated using the aerosol hygroscopicity parameter (Petters and Kreidenweis, 2007):

$$m = m_w + (m_{aer} - m_w)\left(\kappa \frac{a_w}{1 - a_w} + 1\right)^{-1} \tag{5}$$

where $m$ is the refractive index of the wet particle, $m_w$ is the refractive index of pure water, $m_{aer}$ is the refractive index of the dry aerosol particle, $a_w = \text{RH}/100\%$ is water activity neglecting the Kelvin effect, and $\kappa$ is hygroscopicity parameter. The refractive indices $m = n+ki$ include a real ($n$) and imaginary ($k$) component. Equation (5) is derived by combining Eq. (6) in
Shettle and Fenn (1979) and Eq. (1) in Carrico et al. (2010).

Figure 2 illustrates the change in optical properties with RH as modelled via Eqs. (2)-(5) and using the average size distribution measured by the OPC on 2 August 2022. A few notable trends can be summarised as follows. The backscatter coefficient $\beta_{mie}$ varies ~ ±50% between 0 and 80% RH. However, most of the variability is within ±20%. The function only weakly depends on the assumed hygroscopicity parameter. For real refractive indices $n > 1.5$, the backscatter coefficient can
decrease with increasing RH. Increasing the RH increases this scattering cross-section due to hygroscopic growth. However, the backscatter decreases due to a decrease in the refractive index. For aerosols with a larger refractive index, the latter effect can dominate in the 40-90% RH range. The modelled lidar ratio varies between ~20 sr and 80 sr. Under dry conditions, the main controlling factor of the lidar ratio is the refractive index. Larger values of $n$ amplify backscatter and thus result in a lower lidar ratio. Larger values of $k$ amplify extinction and thus increase the lidar ratio. The lidar ratio can increase up to a
factor of 2 with increasing RH, consistent with previous similar numerical simulations (Ackermann, 1998; Zhang et al., 2022).





The modelled $\beta_{\mathrm{mie}}$ is primarily sensitive to changes in aerosol number concentration and to a lesser extent the shape of the size distribution of particles in the $1 < D < 10$ μm diameter range. Figure 3 shows a statistical analysis of the relationship between the aerosol size distribution and optical properties for 2 August 2022. The assumed refractive index is $m = 1.55 + 0i$. The value was picked arbitrarily due to lack of knowledge of the refractive index of the aerosol and is used for illustration

purposes. The aerosol size distribution shown in Figure 3a (red fitted line) shows a coarse mode with a mode diameter ~1 μm. Figure 3b shows that particles between $2 < D < 6$ μm contribute most of the signal to the total backscatter coefficient. Figure 3c shows the correlation of the integrated number concentration for particles $D > 3$ μm against the total $\beta_{\mathrm{mie}}$. Linear regression of these data ($R^2 = 0.98$) can be used to relate an observed $\beta$ to aerosol number concentration. The regression yields an intercept value that corresponds to the $\beta_{\mathrm{mie}}$ that is not explained by particles $D > 3$ μm. As indicated in Figure 2a, the modelled $\beta_{\mathrm{mie}}$

depends weakly on RH. This implies a dependence of the regression slope on RH. Indeed, the example in Figure 3d shows a slightly smaller slope for the assumed RH = 80%. The regression analysis can be applied to arbitrary lower size cuts for the integrated number concentration. One would expect the strength of the correlation would decrease if smaller size particles are included. For example, the correlation for number concentration with $D > 0.01$ μm and total $\beta_{\mathrm{mie}}$ is likely small, since Aitken and Accumulation mode particles insignificantly contribute to the backscatter. However, number concentrations within the

coarse mode will be strongly autocorrelated if the shape of the coarse mode distribution is unchanged. In this case, the number of $D > 1$ μm, $D > 3$ μm, and $D > 7$ μm will all yield a strong correlation with the total $\beta_{\mathrm{mie}}$. Figure 3c tests the extent to which the correlation degrades for different lower size thresholds. This example shows $R^2 > 0.75$ even for a lower threshold $D_{\mathrm{lo}} = 0.53$ μm, but essentially no correlation for $D_{\mathrm{lo}} < 0.5$ μm. The degradation of the correlation is evident as larger scatter in Figures 3(e) and (f), compared to Figure 3(a). Although the details change due to day-to-day variability of the shape of the size

distribution, the overall trends in Figures 3(b)-(f) are repeatable. It is proposed that the empirical regressions shown in Figures 3(d)-(f), combined with a sensible $D_{\mathrm{lo}}$ can be used to retrieve aerosol number concentration $N(D > D_{\mathrm{lo}})$ from measured lidar backscatter coefficients.

Optical models like the one given by Eqs. (1)-(5) have been used successfully to relate OPC size distributions and backscatter for lidar returns from polar stratospheric clouds and cirrus clouds (Cairo et al., 2011; Snels et al., 2021). Here,

however, the aerosol refractive index, the aerosol hygroscopicity, the contribution from molecular absorption to extinction, and the particle aspherical shape are unknown. Combined, these uncertainties are too large to rely on the optical model alone to relate observed inverted backscatter to particle number concentration. Instead, this work relies on empirical correlations between the lidar observed attenuated backscatter at $z = 105$ m (the lowest range gate) stratified by relative humidity and surface-based particle number concentration. Note that the two-way attenuation of the backscatter close to the ground is

minimal and attenuated backscatter at $z = 105$ m approximately equals the true backscatter.

Figure 4 summarises the campaign average correlation between the lidar-observed attenuated backscatter at $z = 105$ m and particle number concentrations $D_{\mathrm{lo}} > 0.53$, $D_{\mathrm{lo}} > 1.03$, and $D_{\mathrm{lo}} > 3.25$ μm. The height $z = 105$ m is the lowest range gate to the surface that has complete time coverage above the signal-to-noise ratio. The shown correlations are analogous to those



shown from the model in Figure 3c. For the regression analysis, a lower threshold in number concentration was imposed, $N >$ 2, 0.5, and 0.02 cm$^{-3}$ for $D_{lo} > 0.53$, $> 1.03$, and $D_{lo} > 3.25$ μm, respectively). Below this threshold the correlations are poor and the regression analysis obscured the intercept value. The correlations show the same pattern as the model. An intercept of the regression line for $N = 0$ cm$^{-3}$ indicates the portion of the backscatter that is not explained by particles with $D > D_{lo}$. The

change in the slope with RH is relatively small and shows a slight decrease in backscatter at $50\% < RH < 65\%$, which is qualitatively comparable to the trends in Figure 2a. The Pearson correlation coefficients are similar for the three shown cutoff sizes. Moving the lower size threshold to the lowest diameter measured by the OPC ($D_{lo} > 0.28$ μm) results in poor correlations ($R^2 < 0.2$) for all RHs. This consistency with the model simulations in Figure 3d is satisfying. The $R^2$ values decrease with increasing RH. At RH > 90%, the correlation is poor ($0.2 < R^2 < 0.5$). This suggests either increasing interference of other

backscattering particles at higher RH or an increasing uncertainty due to uncertainty in the RH value itself. Specifically, the comparisons between the interpolated sonde product and the ground-based meteorological station suggest that the precision of the interpolated sonde RH is not better than ±7% in absolute RH units (supporting information). Furthermore, the $R^2$ values decrease with height, suggesting either interference from backscatter attenuation and/or decorrelation of the aerosol at height $z$ with those at the surface. In summary, the results in Figure 4 suggest that backscatter fluctuations are indicative of particle

number concentration if backscatter exceeds the value of theintercept, if a threshold $D_{lo} > 0.53$ μm is selected, and if RH < 90%.

### 2.4.3 Derivation of Backscatter Flux

Backscatter flux is obtained using the eddy covariance technique,

$$F_\beta = \langle w'\beta' \rangle \tag{6}$$

where $w'$ and $\beta'$ are the instantaneous fluctuations of vertical velocity and attenuated backscatter and the $<>$ indicates the time

average. The measured backscatter coefficients are thresholded at a signal-to-noise ratio of -17 dB, which corresponds to a velocity precision of 20 cm s$^{-1}$ (Newsom and Krishnamurthy, 2020). The backscatter coefficient data shows occasional spikes, possibly due to the transit of larger objects through the beam including birds. These spikes were removed using the following despiking algorithm. A low-pass filter with a cutoff frequency of 0.01 Hz and a 4$^{th}$-order Butterworth filter function is applied to the backscatter coefficient. Values outside the 0.01 and 0.99 quantiles of the ratio of filtered and measured backscatter are

considered spikes and replaced with the value from the filtered data, which approximately corresponds to the average ±100 s of the removed data point. This ensures continuity in the data during the spike event.

The Doppler lidar operates in fixed point vertical orientation in contiguous blocks ~780 s in length with vertical velocity and backscatter reported at ~1 Hz frequency. This is followed by a ~120 s breaks while the instrument plan position indicator scans. The temporal spacing between consecutive timestamps is $1.025 \pm 0.15$ s. Here each contiguous block is used

to derive a particle flux. First, the despiked backscatter data were detrended using a linear fit (Behrendt et al., 2020). Despiking removes spurious peaks from the data, while detrending both subtracts the mean and possible linear trends from the timeseries.





Both are needed to compute accurate fluxes. Figure 5 shows an example of a contiguous block showing detrended and despiked vertical velocity and backscatter data. The corresponding signal variances are $\sigma^2_w = 0.79$ m$^2$ s$^{-2}$ and $\sigma^2_\beta = 0.093$ Mm$^{-2}$ sr$^{-2}$, respectively. The calculated backscatter flux $F_\beta = <w'\beta'> = 1.6\times10^{-5}$ s$^{-1}$ sr$^{-1}$.

The lidar vertical velocity and backscatter data contain uncorrelated random noise stemming from a finite number of scatterers in the sampling volume and low photon counting statistics (Lenschow et al., 2000). The variance from uncorrelated noise can be separated from the signal using various methods. One common approach is the autocovariance method (Lenschow et al., 2000; Wulfmeyer et al., 2016). First, the autocovariance function $A_x(\tau)$ of the time series is computed via:

$$A_x(\tau) = \text{cov}(x_t, x_{t+\tau}) \tag{7}$$

where $x_t$ is the timeseries of interest, $\tau$ is the lag time, and cov is the covariance function. $A_x(0)$ equals the variance of $x_t$. Next $A_x(\tau)$ from the data is fit to a model of the form:

$$A_{model}(\tau) = \nu - k\tau^{(2/3)} \tag{8}$$

where $v$ and $k$ are fitted parameters. The fit is obtained for lags up to the first zero crossing of $A_x(\tau)$. The model extrapolated to zero lag $A_{model}(0)$ equals the noise free variance. The variance attributed to noise is:

$$\delta^2_x = A_x(0) - A_{model}(0) \tag{9}$$

Finally, the integral time scale, $I$, is obtained from the fit parameters via:

$$I = \frac{2}{5}\left(\frac{\nu}{k}\right)^{3/2} \tag{10}$$

Figure 6 shows an example of the autocovariance method applied to the contiguous data block shown in Figure 5. The autocovariance (Eq. (7), black lines) is largest for lag zero and decreases for larger lag times. The model (Eq. (8), red lines)

shows that the decrease with increasing lag is well-modelled using the ⅔ power law relationship. The derived noise variances, taken from the difference between the calculated covariance and model at lag time zero (Eq. 9) are $\delta^2 = 0.161$ m$^2$ s$^{-2}$ and $\delta^2 = 0.052$ Mm$^{-2}$ sr$^{-2}$ for vertical velocity and backscatter data, respectively. The integral time scales derived from Eq. (10) are $I = 21$ and $I = 24$ s, for vertical velocity and backscatter data, respectively. The contribution of the noise variance to the total variance, evaluated as $\delta^2/\sigma^2$ is ~0.2 and ~0.56 for the vertical velocity and backscatter data, respectively. This indicates that

significant noise is present in the data. The derived integral time scales for the two series are similar. For lags > ~50 s, the autocorrelation for both time series is minimal. The example in Figure 6 illustrates how $\delta^2$ and $I$ were determined for each contiguous flux segment.

Spectral analysis is used to estimate the frequency where noise overwhelms the signal. Figure 7 shows the spectral decomposition of the vertical velocity and backscatter data. The spectra $S(\beta)$ are flat for frequencies larger than 0.035 Hz, which indicates white noise. Integrating $\int S(\beta)df$ from 0.035 Hz to the Nyquist frequency equals the noise variance $\delta^2_\beta$ derived





from the autocovariance analysis. Similarly, integrating $\int S(w)df$ from 0.05 Hz to the Nyquist frequency equals the noise variance $\delta^2_w$. These thresholds are similar for other contiguous segments. This suggests that derived backscatter fluxes for frequencies > ~0.035 Hz are not well resolved due to noise in the lidar signal.

### 2.4.4 Quality Control and Uncertainties

*1. Random noise error.* The random noise error in the flux $F_\beta$ is approximated for each contiguous segment using Eq. A22 in Wulfmeyer et al. (2016)

$$\sigma_{noise} \simeq \sqrt{<\beta'^{\,2}> \frac{\delta^2_w}{N} + <w'^{\,2}> \frac{\delta^2_\beta}{N}}$$

(11)

where $\sigma_{noise}$ is the uncertainty in the flux due to random noise, and $\delta^2_w$ and $\delta^2_\beta$ are noise variances derived from the autocovariance analysis.

*2. Limit of detection (LOD).* An alternative method to identify fluxes that are dominated by noise is the lag method (Spirig et al., 2005; Emerson et al., 2018; Islam et al., 2022). First, a lag is applied to the vertical velocity data. If the lag is sufficiently large, the computed $<w'\,\beta'>$ has contributions only from statistical noise. This can be taken as the limit of detection for the flux. The lag should be chosen to be sufficiently large to ensure that the autocovariance is zero, i.e., some multiple of the integral time scale. The median value of the integral time scale of the flux was ~11 s and the 99 percentile is 63 s. We consider

a lag of 200 s sufficient to evaluate the limit of detection using this method.

*3. Sample statistics.* The error associated with the limited number of eddies sampled within a flux segment can be estimated using Eq. (9) In Lareau (2020).

$$\sigma_{sample} = 2\frac{I_{F\beta}}{T}\left[F_\beta - \left(\sigma^2_w - \delta^2_w\right)\left(\sigma^2_\beta - \delta^2_\beta\right)\right]$$

(12)

where $\sigma_{sample}$ is the uncertainty due to sample statistics, $I_{F\beta}$ is the integral time scale of the flux, $T$ is the sampling period (780

s), $\sigma^2_w$ and $\sigma^2_\beta$ are the total variances of vertical velocity and backscatter, respectively, and $\delta^2_w$ and $\delta^2_\beta$ are the noise variances of vertical velocity and backscatter, respectively. Note that the integral time scale of the individual parameters w and $\beta$ are similar (mean $I \sim 22$ and 26 s, respectively), but the integral time scale of the flux $<w'\beta'>$ is shorter (mean $I \sim 11$ s).



*4. Deviation from Ensemble average.*

The flux error due to the departure of the observation from the domain-averaged flux is estimated via (Lareau, 2020; Lenschow et al., 1994)

$$\sigma_{ensemble} \simeq 2 \frac{I_{F\beta}}{T} F_\beta \tag{13}$$

*5. Flux loss correction.* Horst (1997) proposed correcting fluxes for sensors with reduced frequency response by applying a cospectral transfer function to the Ogive of the kinematic heat flux, computing the flux loss, and then correcting the observed flux for the missing contribution. This correction is built on the assumption that cospectra for fluxes of different scalar quantities are similar.

$$F_{corr} = F_\beta \left[ 1 + \left( 2\pi n_m \frac{\tau_c <u>}{z_m} \right)^\alpha \right] \tag{14}$$

where $F_\beta$ is the measured flux, $n_m = 0.085$ and $\alpha = ⅞$ are constants for neutral and unstable conditions, $\tau_c$ is the characteristic time constant of the sensor, $<u>$ is the mean wind speed at the sample height, and $z_m$ is the measurement height. For stable stratification, $\alpha = 1$ and $n_m = 2 - 1.915/(1 + 0.5\ z/L)$ where $z/L$ is the stability parameter, z is the measurement height, and $L$ is the Obukhov stability length calculated from the surface ECOR data using the expression in Launiainen (1995). For unstable conditions $z/L < 0$. Figure 7 and similar analysis for other flux segments suggests that the cutoff frequency for instrument noise is ~0.035 Hz. The relationship between the 3 dB cut-off frequency and the 90/10 response time of an instrument is estimated from the resistor–capacitor circuit analog of a low-pass filter. In that system the characteristic response time is given by a simple analytical solution $\tau_c = 0.35/f$, where $f$ is the 3 dB cut-off frequency (Andrews, 1999). Using 0.035 Hz as cutoff frequency yields as estimate of $\tau_c \sim 10$ s. For unstable conditions, typical ratios for $F_{corr}/F_\beta$ are ~1.3-1.5.

*6. Stationarity.* A key assumption underlying the measurement of eddy-correlation flux is stationarity. Strictly, stationarity implies that the temporal derivatives of the mean field approaches zero during flux detection period, i.e. $dw/dt = 0$, $dT/dt = 0$, and $d\beta/dt = 0$. Stationarity is rarely fully satisfied. Here we use the following metric to characterise stationarity (Foken and Wichura, 1996). The flux segment is divided into 5 min intervals. Then the relative difference between the flux of the entire segment and the mean of the fluxes from the 5-min flux legs is evaluated. Values less than 30% are stationary. The stationarity metric, hereafter referred to as ξ, is reported alongside the $<w'\beta'>$ data.

*7. Turbulence intensity.* Reduced turbulence causing limited air mixing may result in a low bias of retrieved fluxes. At surface sites, the friction velocity u* can be used to identify periods of limited air mixing (Papale et al., 2006; Barr et al., 2013). It is unclear if a u* criterion can be applied to lidar data at $z = 105$. We, therefore, evaluated the turbulent kinetic energy (TKE) at



$z$ = 105 m from the co-located hemispheric scanning Doppler lidar. TKE can be used as a screening criterion to determine if the sample is inside the turbulent mixed layer (Vakkari et al., 2015). We use a criterion of TKE < $10^{-5}$ cm$^{-2}$ s$^{-3}$ to flag periods where reduced turbulence may *potentially* have biassed the flux. The threshold was picked qualitatively as an indication of the presence of turbulence based on Figure 4a in Vakkari et al. (2015). As will be discussed later, our conclusions are not sensitive to this threshold.

*8. Precipitation.* Precipitation may bias the measured Doppler velocity from lidar and can produce spurious backscatter returns (Aoki et al., 2016). Some of these may be removed by the despiking algorithm. Nevertheless, the flux data where any precipitation was measured during a flux segment were flagged and removed in subsequent data analysis.

*9. Flux Footprint.* The flux footprint parameterization by Kljun et al. (2015) was used to calculate the footprint. An advantage of this model is that it can be used outside surface layer conditions and for non-Gaussian turbulence. Variations in lateral velocity and turbulence fluxes are considered. This parameterization is suitable for measurement heights > 20 m. Friction velocity used in this parameterization which was obtained from the scanning doppler lidar deployed at TRACER. Planetary Boundary Layer height was obtained from the North American Mesoscale Forecast System (NAM) model. (Coniglio et al., 2013) compared different PBL schemes with radiosonde observations and came up with the conclusion that the NAM model produced the smallest mean absolute error. Hence the NAM model was used in this study. The surface temperature from the surface meteorological station at the site was used as the temperature at $z$ = 105 m due to lack of high-resolution temperature data at that height.

**2.4.5 Example Backscatter Flux Data**

Figure 8 demonstrates the application of the various data quality control and uncertainties metrics. Figure 8a shows a three-day timeseries of the noise-thresholded and despiked backscatter curtain for broader context. The backscatter shows values more than 100 Mm$^{-1}$ sr$^{-1}$ during nighttime UTC at z ~400 m (pink colours), likely due to the presence of low clouds. Since our analysis focuses on $z$ = 105 only, no additional cloud screening was performed on the dataset. Backscatter signals up to 1500 m are retrieved at ~noon local time. Figure 8b shows the derived backscatter flux with stable conditions identified via $z/L > 0$ being greyed out. Superimposed in red is the limit of detection (LOD) derived for each contiguous segment using the lag analysis. The observed fluxes significantly exceed the values from the LOD analysis. Note that the LOD fluxes can be positive or negative. In contrast, the observed $F_\beta$ values are all positive (indicating particle emission from the surface). Figure 8c shows that most flux values are from data that pass the stationarity test, i.e. abs($\xi$) < 0.3. Figure 8d shows flux loss corrected hourly averaged flux values during unstable conditions only. The grey shading comprises the cumulative uncertainty of $\sigma_{noise}$, $\sigma_{sample}$, and $\sigma_{ensemble}$ and remains small relative to the absolute value of the retrieved backscatter flux. Taking the combined data quality and uncertainty metrics into account, the data in Figure 8 shows that the backscatter fluxes are statistically significant.



## 2.4.6 Number Flux from Backscatter Flux

Fluctuations in elastic backscatter correspond to fluctuations in aerosol number concentration (Pal et al., 2010). As demonstrated in Figs. 3 and 4, the Doppler lidar is sensitive to particle number concentration $D > \sim0.5$ μm. The relationship between backscatter and particle number is:

$$\beta(S) = \left(\frac{\partial \beta}{\partial N}\right)_S N(S) + c(S) \tag{15}$$

where $S$ is the saturation ratio ($S = RH/100\%$), $\beta(S)$ is the backscatter at saturation ratio $S$, $N(S)$ is the number concentration of particles larger than a specified threshold diameter, $(\partial \beta/\partial N)_S$ is the slope and $c(S)$ is the intercept of the regression lines shown in Figures 4a-l. In practice, $(\partial \beta/\partial N)_S$ and $c(S)$ were empirically evaluated in intervals $[S;S +0.05]$. The ambient $S$ value is obtained at the lidar height and sample time from the interpolated sonde product. The closest calibrated slopes and intercepts are used to derive $N(S)$ from the observed $\beta(S)$.

Figure 9 summarises the retrieved particle number concentration from lidar backscatter via Eq. (15). Here retrievals are limited to conditions where RH < 90%, there is no precipitation, and where the observed backscatter exceeds 1.5 times the intercept value in Eq. (15). The latter limit is necessary to avoid retrievals where the backscatter is dominated by signals unrelated to particle number concentration. The cutoff value was selected to filter most noise while maintaining sufficient signal coverage. Increasing the value to > 1.5 does not further improve the accuracy of the comparison between the lidar and
OPC shown in Figure 9. The lidar retrieved concentrations are visibly noisier, which is likely due to the noise in the backscatter data (Figs. 6 and 7), and the fact that the calibrated slopes were obtained from the campaign average, thus not accounting for variations in refractive index and hygroscopicity in the retrieval. Nevertheless, the retrieval captures the broad trends in particle number concentration, including the transition from lower concentration to higher concentration periods 11 June, 15 July, and 20 July. Pearson correlation coefficients between the OPC and lidar-derived number concentrations are $R^2 = 0.74$ for the time
series shown in Figures 9 (c)-(e). The $R^2$ values are not identical but very similar, which is explained by the autocorrelation of number concentration within the coarse mode, i.e., because shape of the coarse mode is approximately constant throughout the campaign. Overall, the results in Figure 9 confirm that the variability in backscatter is related to changes in coarse mode particle number concentration.

     Based on the preceding paragraph, we assert that high-frequency fluctuations in backscatter are related to high-
frequency fluctuations in number concentration. To obtain the number flux from backscatter flux, high frequency fluctuations in relative humidity also need to be considered (Fairall, 1984). This is because hygroscopic growth increases the particle size which in turn affects the backscatter. During periods of intense sensible and latent heat flux, there are systematic differences in relative humidity or saturation ratio in updrafts and downdrafts, resulting in a saturation ratio flux. This saturation ratio flux can lead to an apparent particle flux that is not related to turbulent transport. Thus, turbulent flux measurements require





correction for false fluxes during periods of high saturation ratio flux (Fairall, 1984; Kowalski, 2001; Vong et al., 2004; Islam et al., 2022).

To derive the influence of saturation ratio fluctuations on the lidar-derived number flux, we use an equation analogous to Eq. (18) in Fairall (1984):

$$\beta' = \left(\frac{\partial \beta}{\partial N}\right)_S N' + \left(\frac{\partial \beta}{\partial S}\right)_N S' \tag{16}$$

Therefore, the number flux can be written as

$$\langle w'N' \rangle = \langle w'\beta' \rangle \Big/ \left(\frac{\partial \beta}{\partial N}\right)_S - \left(\frac{\partial \beta}{\partial S}\right)_N \Big/ \left(\frac{\partial \beta}{\partial N}\right)_S \langle w'S' \rangle \tag{17}$$

where $\langle w'N' \rangle$ is the eddy-covariance flux of particles and $\langle w'S' \rangle$ is the saturation ratio flux. Evaluation of the terms $(\partial \beta / \partial S)_N$, $(\partial \beta / \partial N)_S$, and $\langle w'S' \rangle$ is provided in the supporting information. Note that the saturation ratio flux depends on the latent and sensible heat flux and can be either positive or negative. Furthermore, $\langle w'N' \rangle$ is a net flux and includes contributions from surface emissions and dry deposition. The emission flux can be obtained by removing the estimated contribution from particle

dry deposition (Nilsson et al., 2021)

$$F_{emission} = \langle w'N' \rangle + v_d \langle N \rangle \tag{18}$$

where $v_d$ is the dry deposition velocity, and $\langle N \rangle$ is the average number concentration. Note that by our convention positive $\langle w'N' \rangle$ corresponds to an upward flux. In the absence of emissions ($F_{emission} = 0$), the observed $\langle w'N' \rangle$ would be negative (downward), reflecting the dry deposition process, and using the convention that the deposition velocity is a positive number.

Deposition velocity is a function of particle diameter and land use type. Precise values remain poorly constrained within data showing approximately two orders of magnitude in scatter (Emerson et al., 2020). Here we consider $v_d = 1$ cm s$^{-1}$ as an approximate upper bound for particles in the 0.5 to 5 µm size range depositing on a grassland surface (Emerson et al., 2020).

Combining all the correction included in Eq. (14), Eq. (17), and Eq. (18), the emission flux can be conceptually decomposed into four terms:

$$F_{emission} = F + F_{flc} + F_{wS} + F_{dep} \tag{19}$$

where $F = \langle w'\beta' \rangle / (\partial \beta / \partial N)_S$ is the first order conversion from backscatter to number flux and $F_{flc}$ is the additional flux computed from the flux loss correction due to low frequency response of the Doppler lidar (Section 2.4). If no correction is required $F_{flc} = 0$. The sign of $F_{flc}$ is the same sign as $F$. The term $F_{wS}$ is the apparent contribution to the flux due to variation in the saturation ratio in updrafts and downdrafts. $F_{wS} = - (\partial \beta / \partial S)_N / (\partial \beta / \partial N)_S \langle w'S' \rangle$ and can be either positive or negative. The term $F_{dep} = v_d \langle N \rangle$ is always positive.



## 3. Results

Figure 10 shows a timeseries of the daily averaged lidar retrieved emission flux for particles $D > 0.53$ μm. Only contiguous flux segments exceeding the limit of detection during unstable conditions are included. This excludes night-time periods. Furthermore, flux legs with precipitation present and flux legs where turbulent kinetic energy was below $10^{-5}$ cm$^{-2}$ s$^{-3}$ were

excluded. Increasing the turbulent kinetic energy threshold filters more data, but doesn't alter the overall trends shown in Figure 10. Averaging over the entire day reduces the random errors from noise and short sampling periods (Figure 8) and thus those errors are not further considered here. Several days show missing fluxes (white areas). These are predominantly from days where all the $<w'\beta'>$ fluxes were below the limit of detection. All of the base flux values ($F$, black colour), which are derived from the backscatter flux without further correction, are positive. This suggests that the site is dominated by emissions.

Applying the flux loss correction (gold colours), which accounts for the reduced frequency response of the lidar, increases the base flux by ~30-50%. The correction for saturation ratio flux ($F_{wS}$) increases the flux further. The systematic increase is because on average $<w'S'>$ is negative during the daytime. However, the correction is small. The correction shown for dry deposition is based on an assumed $v_d = 1$ cm s$^{-1}$. In general, this correction is small relative to the reported emission flux values. However, the contribution may be appreciable during high concentration periods. Nevertheless, the red band in Figure 10 is

likely an overestimate, as the true deposition velocity is likely an order of magnitude, and perhaps two orders of magnitude lower than the assumed $v_d = 1$ cm s$^{-1}$ in Figure 10 (Emerson et al., 2020).

The emission fluxes in Figure 10 range from ~10 cm$^{-2}$ s$^{-1}$ to ~100 cm$^{-2}$ s$^{-1}$. For an assumed boundary layer height of 1 km, and a day length of 100,000 s, a sustained flux of 10 cm$^{-2}$ s$^{-1}$ corresponds to an increase in number concentration of 10 cm$^{-3}$ throughout the boundary layer due to the emission flux. Thus, the emission is significant relative to the background

concentration, which varies between 1 and 20 cm$^{-3}$ (Figure 9c). Timeseries like Figure 10 can be created for particle fluxes $D > 1$ μm and $D > 3$ μm. The temporal trend is identical to the data shown in Figure 10 because they are scaled to the same backscatter flux. However, the magnitudes of the flux values are reduced to 33% and 1.5% of the emission flux for particles $D > 0.53$ μm, respectively. The flux footprint evaluated only for included flux segments (i.e. unstable conditions) is $3.1 \pm 2.4$ km (mean $\pm$ 1 standard deviation) and includes a mix of grassland fields, asphalted surfaces, and urban housing developments.

The footprint in Figure 1 is an illustrative example of a simulated 2D footprint at the site. The timeseries show significant autocorrelation, with multi-day periods of higher fluxes, followed by multi-day periods of lower fluxes. This begs the question. What are the sources of the emissions?

Possible candidate sources include dust emitted from the soil or biological particles emitted from vegetation. These sources would be expected to scale with meteorological conditions. For example, wind-blown soil dust would be expected to

correlate with friction velocity (Kok et al., 2012). Biological emissions might be expected to respond to relative humidity (Wright et al., 2014; Yadav et al., 2022), with higher relative humidity triggering emission. Due to the sustained emission of coarse particles over long periods of time, it seems unlikely that anthropogenic point sources account for the emission. For



example, hypothetical activities like traffic, airport landings and takeoffs, etc would be expected to have a punctuated signal in the observed time series of particle flux. These would manifest in peaks with, for example, rush hour or weekday/weekend cycles. No such periodicity is apparent in the high time resolution flux data (not shown here). Natural sources, therefore, appear to be the most likely candidate.

5       Exploratory statistical analysis was performed by visually examining scatterplots between the observed daily averaged emission flux and potential explanatory variables, including wind speed, wind direction, friction velocity, surface temperature, surface relative humidity, latent heat flux, sensible heat flux, flux footprint, Monin Obukhov length, and soil moisture (Supporting Information). Two candidate variables were identified from this statistical analysis to result in strong emissions: low surface relative humidity and high wind speed/friction velocity. Note that the fluxes are obtained during

daytime when the relative humidity is generally lowest due to surface heating. Further note that the Pearson correlation coefficient for a linear relationship between friction velocity and flux is poor. However, the relationship is fully consistent with that of a power law.

        Figure 11 summarises the dependency of the daily averaged emission flux with surface-derived friction velocity. In this analysis, only the first three terms on the right-hand side of Eq. (19) are included. Corrections for the deposition flux are

not considered, because the value is negligible for the expected typical value of $v_{\mathrm{d}}$ = 0.1 cm/s. The largest observed emission flux coincides with high friction velocity and low relative humidity. The emission flux scales approximately with $u*^4$, which is consistent with wind-blown dust emissions (e.g., Figure 37 and discussion in Kok et al., 2012). Indeed, the emission flux can be parameterized as $F = 3000u*^4$ where $u*$ is the friction velocity in m s$^{-1}$ and the emission flux $F$ is in cm$^{-2}$ s$^{-1}$. It is important to note that the comparison to wind-blown desert dust is only an analogy, as the emission mechanisms are likely

different from the saltation process occurring in sandy areas. We further note that no relationship was found between fluxes and soil moisture. Thus, the anticorrelation with relative humidity may or may not be coincidental. It is possible that days with sustained winds also had low relative humidity. However, it is also plausible that low relative humidity aided the lofting of the dust, especially from dust situated on asphalted and other urban surfaces. Longer-term studies will be needed to understand the emission mechanisms and important parameters that influence emissions.

**4. Discussion and Implications**

        This work has laid out a new methodology on how coherent Doppler lidar data can be used to obtain emission number fluxes. First, it was shown that the Doppler lidar can be used to measure backscatter flux using the eddy-covariance technique. Implementing the current state-of-the-science uncertainty analyses demonstrated that the backscatter fluxes are statistically significant (Figure 8). Random errors due to signal noise were present but did not dominate the signal. Next, it was shown that

lidar attenuated backscatter at $z$ = 105 m can be related to particle number concentration above a size threshold using an empirical calibration between attenuated backscatter stratified by relative humidity and particle number concentration above



a certain size threshold measured by an optical particle counter at the surface. Three thresholds were considered: $D > 0.53$ μm, $D > 1$ μm and $D > 3$ μm. It was shown that this calibration led to a good correlation between particle number concentration inferred from lidar backscatter and particle number concentration observed at the surface (Figure 9). This calibration was applied to derive the particle number emission fluxes from the backscatter flux, including flux loss correction for bias from

reduced frequency response by the lidar (Figure 7 and Section 2, Horst 1997), correction for bias due to saturation ratio flux (Fairall 1983), and correction for bias due to particle deposition (Nilssen et al., 2021). The magnitude of these corrections was shown to be appreciable but not dominant. The largest of these corrections was the flux loss correction due to reduced frequency response, which led to an ~30-50% reduction in the observed emission flux (Figure 10). The temporal trend of the flux showed strong autocorrelation with multi-day periods of higher followed by multi-day periods of lower emission fluxes.

The emission flux was correlated with wind speed/friction velocity and anticorrelated with surface relative humidity (Figure 11). This suggests that the emissions are due to mechanical erosion from urban surfaces.

   We are not aware of other coarse mode particle flux measurements in urban environments. To place these emission data in context, we compare the emission flux reported here to other field measurements of emission fluxes from eroding soils (cf. Figure 37 Kok et al., 2012). Those measurements report mass fluxes varying between 10 and 100,000 μg m$^{-2}$ s$^{-1}$. The

approximate mass for a 1 μm particle (roughly the mass mode diameter of the coarse mode distribution) is ~$10^{-6}$ μg. Applying this factor to the emission number flux in Figure 11, yields emission fluxes between 0.1 and 1 μg m$^{-2}$ s$^{-1}$, which is much lower than the range of dust fluxes (Kok et al., 2012). The composition of the coarse mode in urban environments shows contributions from road dust, tire debris, and biological particles (Wu and Boor, 2021). We would expect that the reported emissions emanate from dust located on asphalted surfaces and exposed soil from grasslands and gardens.

Significant data quality screening and averaging were applied to the dataset here. This led to the exclusion of flux data under stable conditions (Monin-Obukhov length > 0) and in the presence of precipitation. These data are not necessarily bad. However, evaluating data under stable conditions, which predominantly occurred during nighttime and also coincided with low u*/TKE conditions will require more depth analysis to understand the importance and accuracy of the flux loss correction and footprint expansion that occurs in that regime. Furthermore, we mostly considered daily average fluxes. Higher

time resolution data are available (e.g., Figure 8d). These data are noisier (high flux leg-to-flux leg variability) and generally show a diurnal trend with peak fluxes occurring near local noon. Understanding and fully quantifying these higher time resolution emission patterns may be important for understanding the influence of the emissions on cloud formation. The retrievals in this work were limited to a single vertical level at $z = 105$ m. In principle, this method will be suitable to also retrieve flux profiles, i.e., the variation of particle number flux with height. This work did not further pursue this for several

reasons. First, the calibration of the signal against the surface optical particle counter (Figure 4) becomes less certain when applied to backscatter at higher altitudes. This is due to increasing uncertainty due to elevated relative humidity (Figure S1), the potential decoupling of higher layers from the surface observations, and the loss of signal due to the two-way attenuation of the backscatter signal. Those issues can be overcome, at least in principle, by incorporating vertical profile measurements



of particle concentration and by applying appropriate inversion to convert attenuated backscatter to backscatter. Second, measurements at higher altitudes will require careful screening for cloud events, boundary layer height, and boundary layer evolution. This is illustrated in Figure 8a, which shows how clouds, and a low signal-to-noise ratio would complicate automated evaluation of a time series at $z = 1000$ m. In summary, evaluation of vertical profiles may be possible in future studies but will

require additional data and a case-by-case analysis approach. In aggregate, the applied methodology resulted in a first-pass analysis of the data. Some of the assumptions may be relaxed with additional analyses, resulting in additional information that may be derived from this and similar datasets.

        Overall, the results here are promising to obtain emission fluxes using remote sensing techniques. The advantages of this approach are that the technique is relatively low maintenance, thus allowing long-term measurements. The fluxes are

obtained at 100 m or higher. Therefore, a high frequency response instrument sampling at 10 Hz is not as critical, because turbulent eddies are more coherent at higher altitudes. Another advantage is that the flux footprint increases with height, thus allowing the sampling of emissions from a several km$^2$ large area. However, the lidar-derived number fluxes also have several limitations. The lidar backscatter is a convolution integral property e.g., Eq. (3), that is sensitive to relative humidity. The relationship between backscatter and number flux will always be subject to the limitation of the assumptions made in the

analysis. For example, a campaign averaged calibration was used to relate backscatter to particle number, which neglects potential variations in the shape of the coarse mode size distribution, variations in aerosol refractive index, variations in particle shape, and variations in aerosol hygroscopicity. The seriousness of these assumptions is difficult to evaluate, because a large enough dataset is required to build the calibration in Figure 4. Future studies may consider a longer-term dataset, subdivide the calibration into multiple time periods and then evaluate the influence of using different calibrations on the retrieved number

flux. Another limitation is the inherent noisiness of lidar backscatter data due to instrument noise and limited photon backscatter from the control volume. This might limit the application of this technique to regions where sufficient backscatter is available, as defined by a high signal-to-noise ratio. The overall limit of detection appears to exceed values of expected dry deposition fluxes in the sampled size range. For example, the largest estimate for the deposition flux is ~10 cm$^{-2}$ s$^{-1}$ (Figure 10). More realistic values of the dry deposition velocity would result in deposition fluxes of 1 or 0.1 cm$^{-2}$ s$^{-1}$. These values are

unlikely to be resolvable with the Doppler lidar in this configuration and environment. Despite these limitations, the technique may have broad applicability to evaluate surface emissions from deserts, oceans, urban landscapes, and biologically active ecosystems, which all may release appreciable coarse mode particles through wind and weather-driven processes.

        An important implication of these emissions is the role of the coarse mode particles on cloud glaciation through ice nucleating particles. It is well established that ice nucleating particle (INP) number concentrations correlate with particle

number concentration with diameter > 0.5 µm (Georgii, 1959; DeMott et al., 2010). DeMott et al. (2010) give a parameterization that predicts INP concentration as a function of temperature ($T$) and aerosol number concentration ($n_{aer,0.5}$). The parameterization depends nonlinearly on the aerosol concentration. Nevertheless, the particle number flux retrieved here can be used to estimate INP emissions as a function of temperature. Figure 12 shows the calculated INP concentration vs.



temperature curve for an assumed background number concentration $n_{aer,0.5} = 1$ cm$^{-3}$ (black solid line). Active fluxes will increase the $n_{aer,0.5}$ over time, which in turn will increase INP. Here we estimate this increase assuming a 1 km thick boundary layer, no sinks for INP, and emissions that are continuously active for 24 hr duration. Figure 12 shows the increase in INP concentration for three different flux strengths. For the strongest assumed flux (100 cm$^{-2}$ s$^{-1}$), the calculated INP increases by

a factor of ~3, ~10, and ~35 at $T = -10, -20$, and $-30$ °C, respectively. These fractions are sensitive to the assumptions (background concentration, boundary layer height, emission duration, and emission flux). The selected scenario likely presents an upper limit estimate for the expected influence of the emissions on INP concentrations during the TRACER campaign. Estimates such as those shown in Figure 12 may be critical for understanding the emissions of INP and their influence on clouds. However, given the current uncertainties, these estimates will require validation against direct measurements of INP

and the effects may vary on the temperature regime. For example, at $T > -20$ °C, rarer biological particles likely dominate INP composition (Petters and Wright, 2015; Kanji et al., 2017; Cornwell et al., 2022). Those warm INP often correlate with fluorescent properties of particles (Wright et al., 2014; Cornwell et al., 2022). Calibration of the Doppler lidar signal against a fluorescence sensing optical particle counter instead of a generic optical particle counter might provide important insights into this process.

**5. Code availability**

Not applicable

**6. Data availability**

Data sets used here obtained from the Atmospheric Radiation Measurement (ARM) user facility, U.S. Department of Energy (DOE), particularly from the ARM Mobile Facility Houston, TX, all for 06-01-2022 to 08-10-2022, are as follows:

-    Optical Particle Counter (AOSOPC), data set accessed 2022-11-16 at http://dx.doi.org/10.5439/1824224 (Cromwell and Singh, 2021).

     -    Doppler Lidar (DLFPT), data set accessed 2022-11-16 at http://dx.doi.org/10.5439/1025185 (Newsom and Krishnamurthy, 2021).

     -    Interpolated Sonde (INTERPOLATEDSONDE), data set accessed at 2022-12-16 at

25        http://dx.doi.org/10.5439/1095316 (Jensen et al., 2021).

     -    Surface Meteorological Instrumentation (MET), data set accessed 2022-11-16 at http://dx.doi.org/10.5439/1786358 (Kyrouac and Shi, 2021).

     -    Eddy Correlation Flux Measurement System (30ECOR), data set accessed 2022-12-16 at http://dx.doi.org/10.5439/1025039 (Sullivan et al., 2021).



- Doppler Lidar Horizontal Wind Profiles (DLPROFWIND4NEWS). Compiled by T. Shippert, R. Newsom and L. Riihimaki.ARM Data Center. Data set accessed 2022-12-16 at http://dx.doi.org/10.5439/1178582 (Shippert et al., 2022).

## 7. Competing interests

5    The authors declare that they have no conflict of interest.

## 8. Acknowledgements

This work was funded by the DOE Office of Science, Biological and Environment Research, Grant No. DE-SC0021074, National Aeronautics & Space Administration (NASA) Grant No. 80NSSC21K1137, and NCSU NC Space Grant Consortium Grant No. 2022-2155-NCSU.

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



## 10 Figures

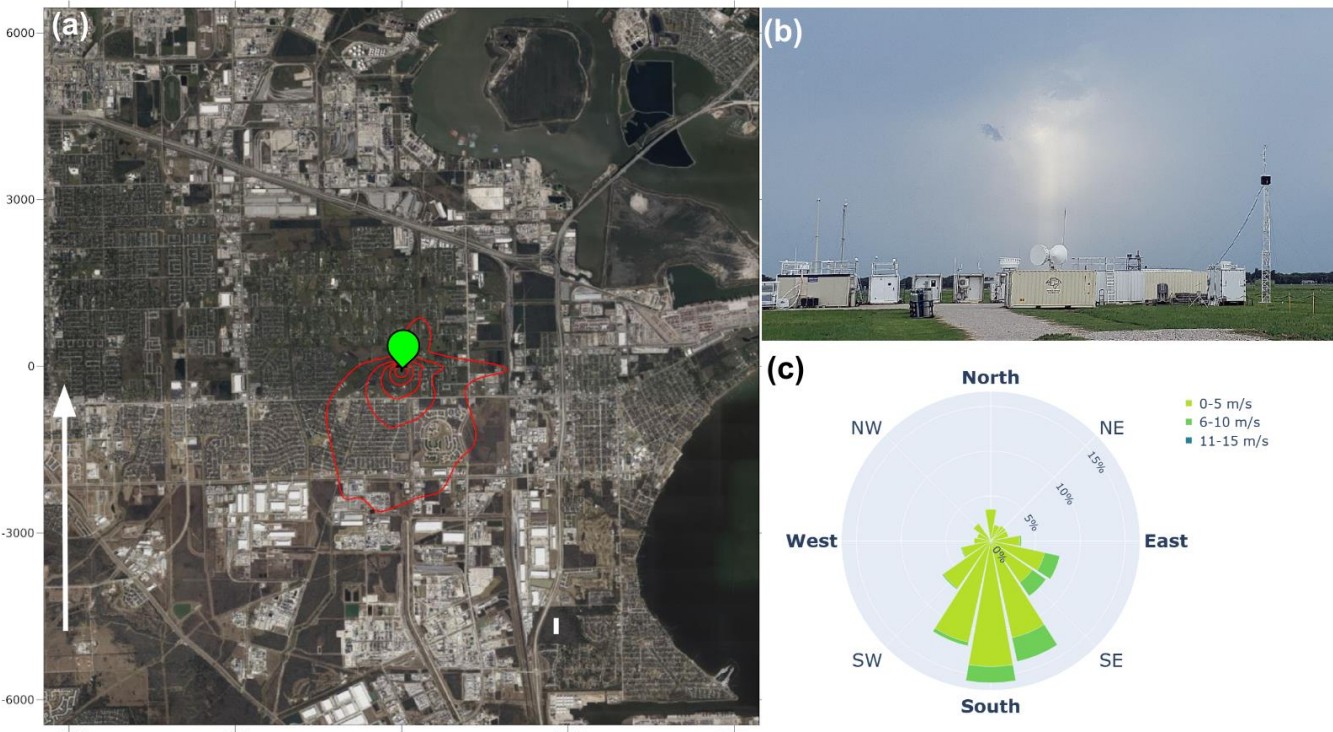

**Figure 1:** (a) The location of the AMF (green location marker) with red polygon represents the flux footprint area. Map data: Google, CNES / Airbus, Houston-Galvaston area council, Landsat/Copernicus, Maxar Technolgies, Texas General Land Office, U.S. Geological Survey, USDA/FPAC/GEO, 2023. The white arrow gives the prevailing wind direction during the campaign. (b) Picture of the AMF setup. (c) Windrose diagram for wind direction and speed during the sampling period.



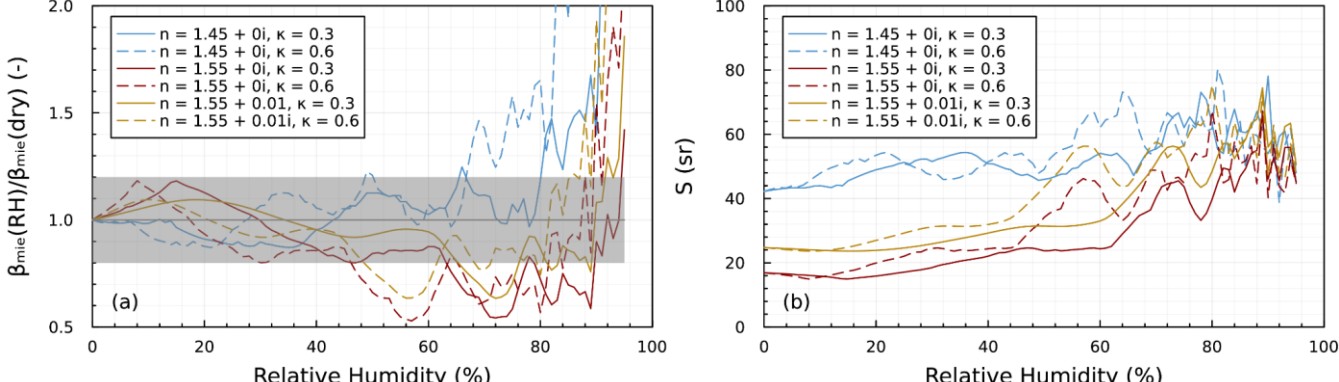

**Figure 2:** Change in aerosol optical properties with relative humidity as a function of refractive index and hygroscopicity. (a) Change in $\beta_{mie}$ as a function of relative humidity. Colour indicates the assumed refractive index. Solid and dashed lines correspond to $\kappa = 0.3$ and $\kappa = 0.6$, respectively. The grey shading indicated ±20% variability. (b) LIDAR ratios as a function of relative humidity.



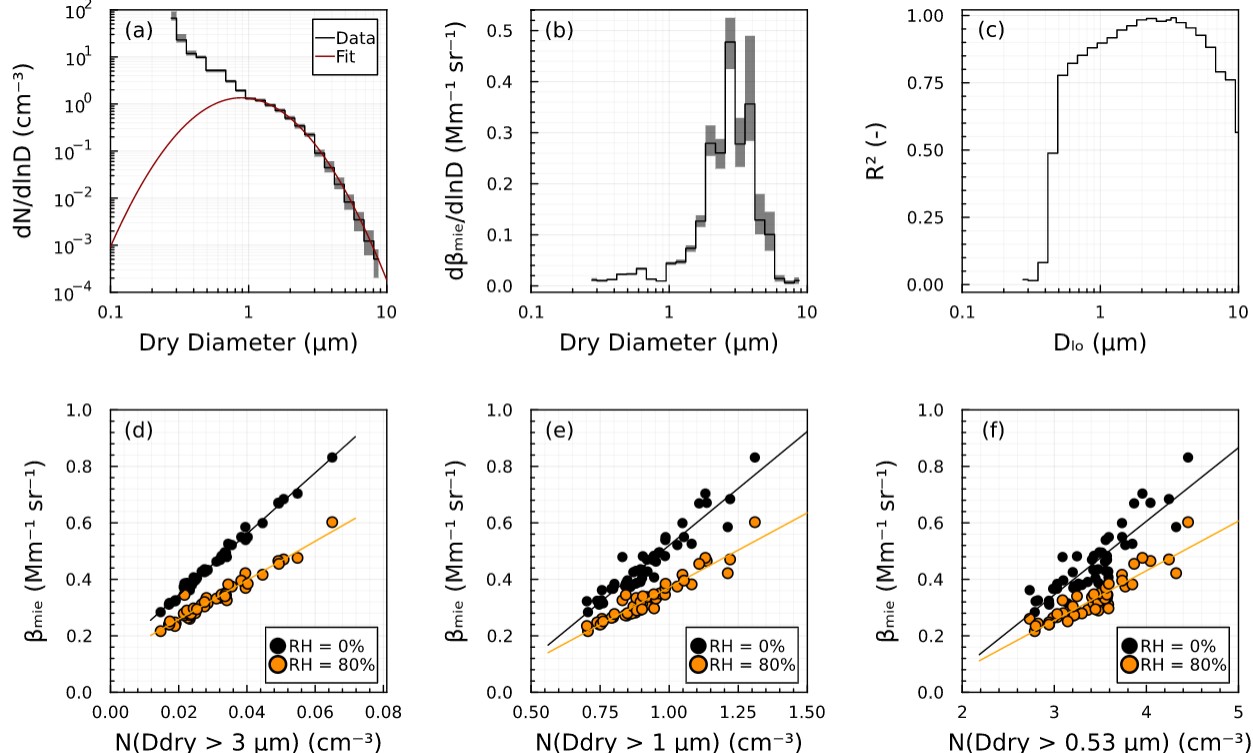

**Figure 3:** Statistical analysis of one day of aerosol size distributions. (a) Average aerosol size distribution measured by the OPC. The grey shading indicates the interquartile range. The red line shows a lognormal fit to the coarse mode. (b) Average size-resolved modelled $\beta_{mie}$, expressed as spectral density. The grey shading indicates the interquartile range. (c) Pearson correlation coefficient between $N(D_{dry} > D_{lo})$, the OPC integrated number of particles with diameter exceeding a lower threshold $D_{lo}$, and $\beta_{mie}$ at RH = 0%. (d) Correlation between integrated $\beta_{mie}$ (all sizes) and measured number concentration > 3 μm. Each point corresponds to a 30 min time average. Solid lines indicate a linear fit. Colours indicate the assumed relative humidity. (e) Same as panel (d) but for number concentration > 1 μm. (f) Same as panel (d) but for number concentration > 0.53 μm.





**Figure 4:** Panels (a)-(l): correlation of aerosol number concentration between the lidar observed backscatter at $z = 105$ m and particle number concentrations $D_{lo} > 3.25$ μm (a)-(d), $D_{lo} > 1.03$ μm (e)-(h), and $D_{lo} > 0.53$ μm (i)-(l). The data are stratified by ambient relative humidity, 45-50% (a), (e), and (i), 55-60% (b), (f), (j), 65-70% (c), (g), and (k), and 75-80% (d), (h), and (l). Panel (m) shows the slope of the regression line as a function of cutoff diameter $D_{lo}$ and relative humidity using intervals $[RH, RH + 5\%]$. Panel (n) shows the Pearson correlation coefficient $R^2$ for the regression as a function of cutoff diameter $D_{lo}$ and relative humidity using intervals [RH, RH + 5%].





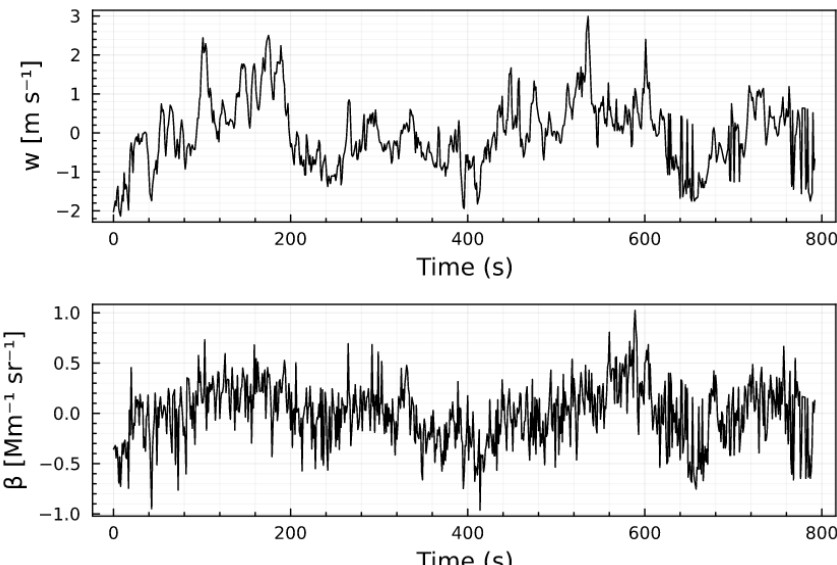

**Figure 5:** Example 780 s contiguous block of (a) vertical velocity and (b) backscatter data.

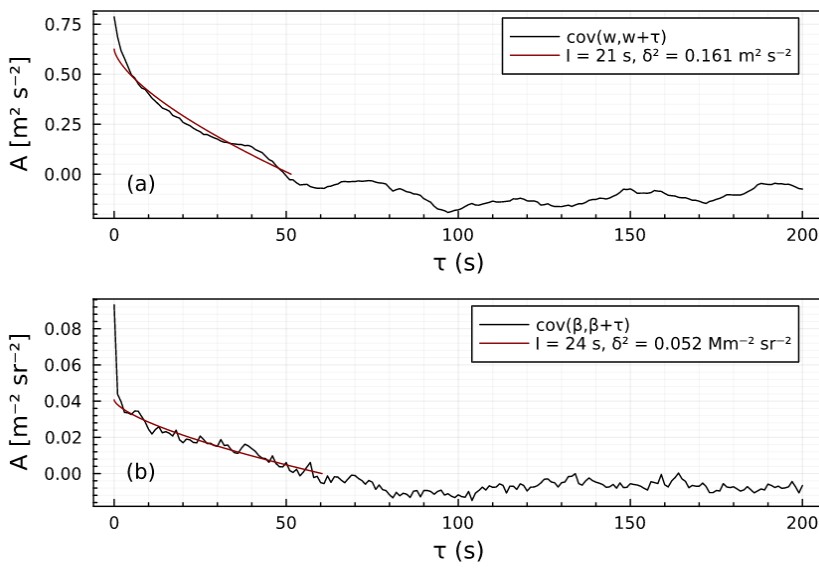

**Figure 6:** Autocovariance function for the data shown in Figure 5. (a) vertical velocity and (b) backscatter data. Black: $A_x(\tau)$, red: $A_{model}(\tau)$.



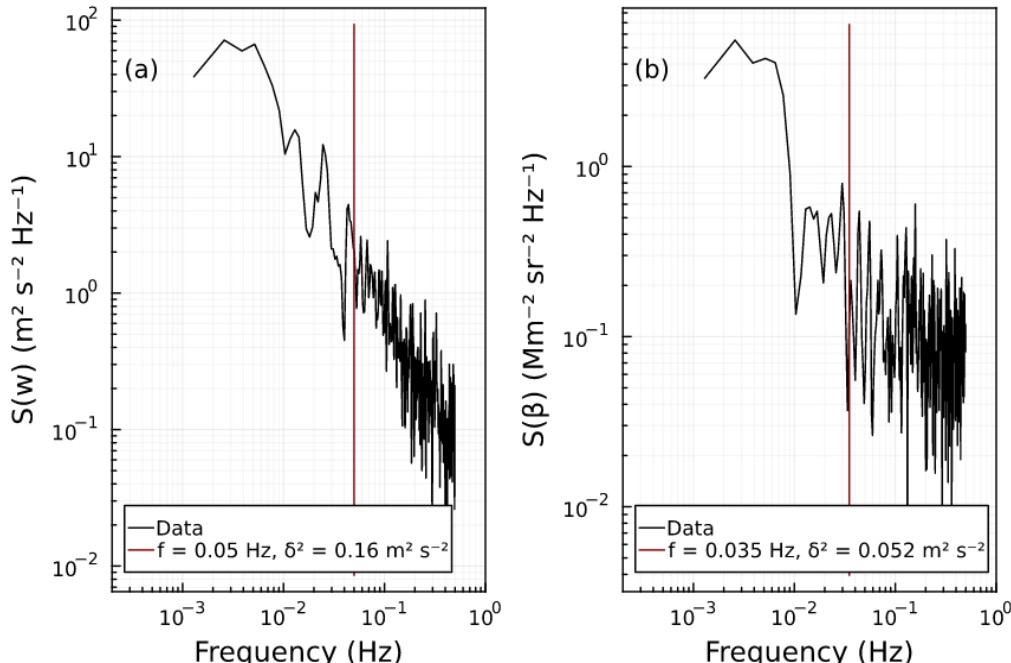

**Figure 7:** Spectral decomposition of the total variance for (a) vertical velocity and (b) backscatter for the contiguous data segment shown in Figure 5. The integral ∫Sdf equals the total variance. The red line corresponds to the frequency cutoff such that the integral from the frequency cutoff to the Nyquist frequency corresponds to the noise variance $\delta^2$ derived from the autocovariance analysis in Figure 6.





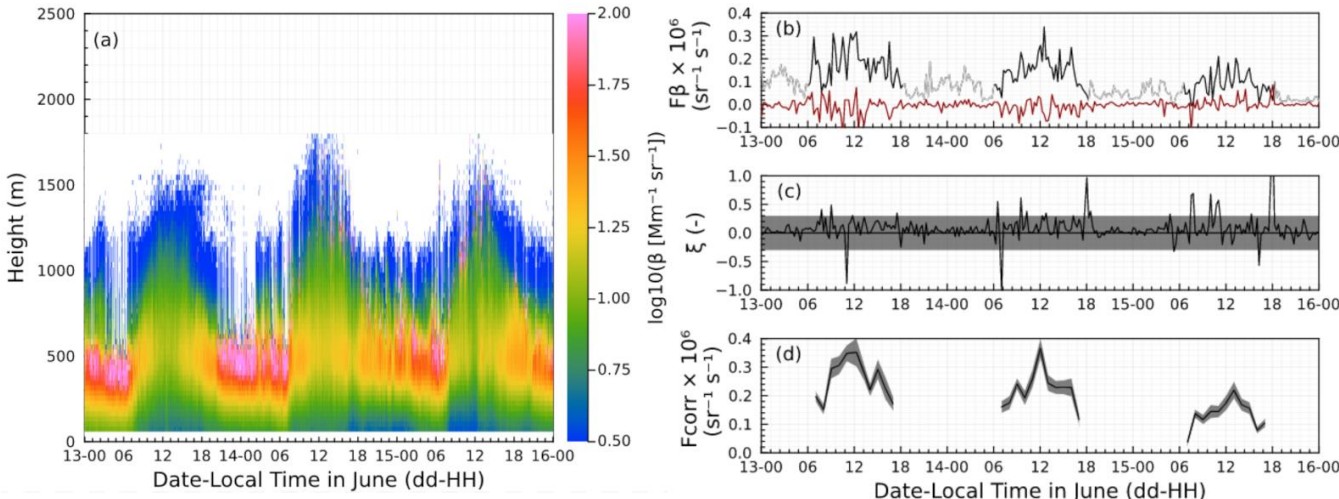

**Figure 8:** Three-day time series starting June-13 2022. (a) Time-height noise-thresholded attenuated backscatter curtain. Colours correspond to the log10 of the backscatter value in units of Mm$^{-1}$ sr$^{-1}$. (b) Black: Backscatter flux at $z = 105$ m for each contiguous segment with $z/L < 0$ (unstable conditions), Grey: backscatter flux at $z = 105$ m for each contiguous segment with $z/L < 0$ (stable conditions). Red: limit of detection (LOD) computed using the lag method. (c) Stationarity metric $\xi$. The grey shading denotes the $\pm 30\%$ threshold given by Foken and Wichura (1996).(d) Hourly averaged fluxes for unstable conditions after application of the flux loss correction. The grey shading indicates the combined uncertainty derived from $\sigma_{noise}$ + $\sigma_{sample}$ + $\sigma_{ensemble}$.



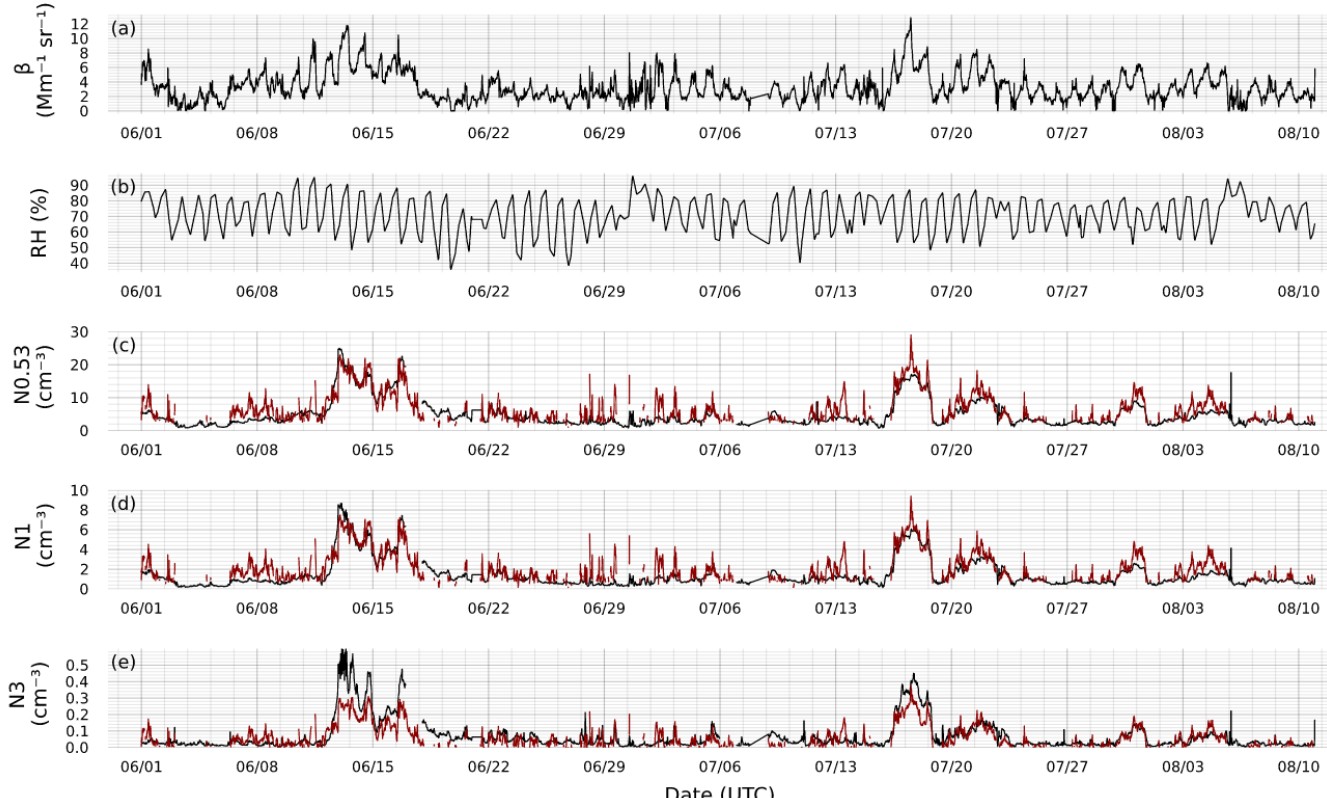

**Figure 9:** (a) Attenuated lidar backscatter at $z = 105$. (c) Relative humidity at $z = 105$ m from the interpolated sonde product. Data are presented at 30 min temporal resolution. (c) Particle number concentration for particle $D > 0.53$ μm measured at the surface by the OPC (black) and retrieved from the backscatter via Eq. (15). (d) Sane as (c) for $D > {\sim}1$ μm size particles. (e) Sane as (c) for $D > {\sim}3$ μm size particles.



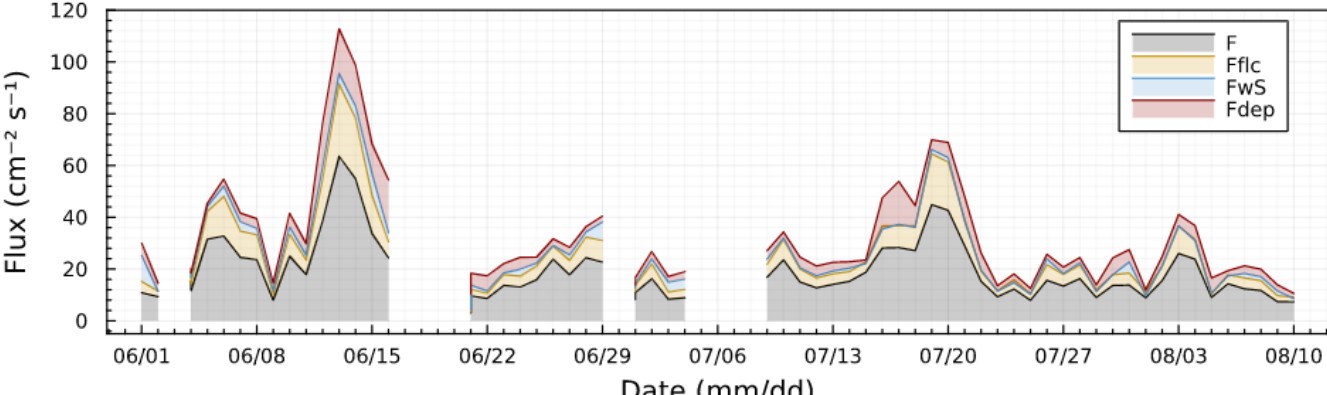

**Figure 10:** Temporal trend of daily averaged lidar retrieved daytime emission flux for particles $D > 0.53$ μm. Colours correspond to the contributions to the emission flux as given in Eq. (19). White areas correspond to dates where fluxes were below the detection limit or insufficient data were available to compute flux corrections.



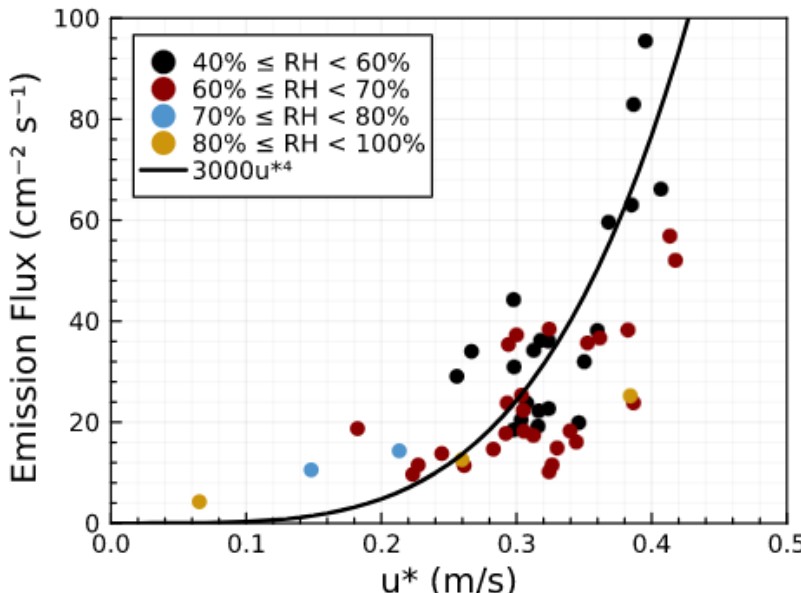

**Figure 11:** Variation of the daily averaged emission flux with surface-derived friction velocity. Colours show data stratification by relative humidity. The solid line illustrates an example of a power law dependency of the emission flux.

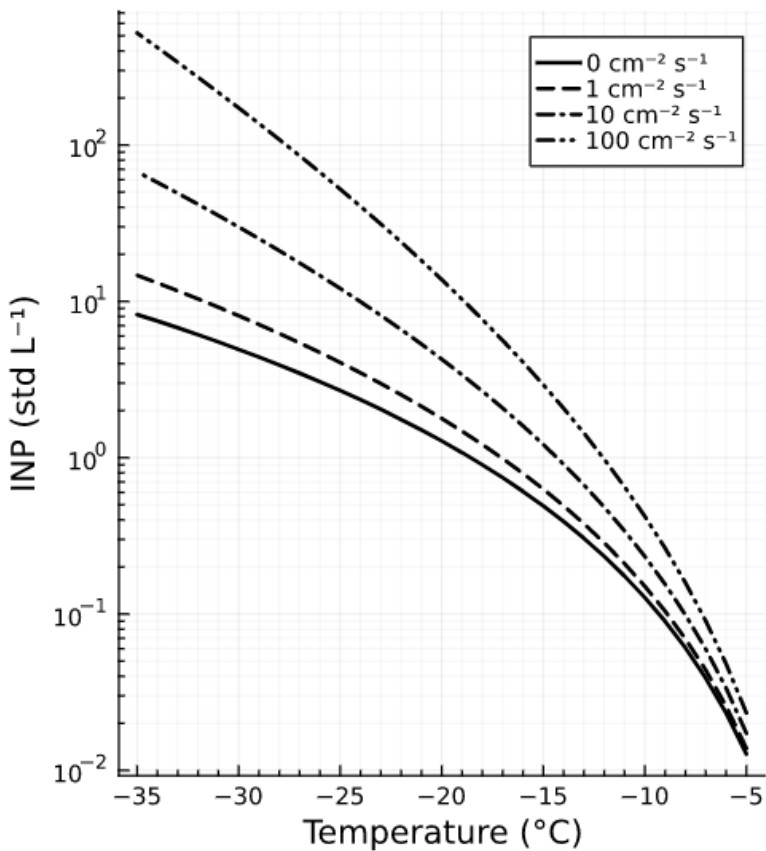

**Figure 12:** Estimated ice nucleating particle number concentration on the DeMott et al. (2010) parameterization for an assumed background number concentration $n_{aer,0.5} = 1$ cm$^{-3}$, assumed aerosol number fluxes for particles $F = 0, 1, 10, 100$ cm$^{-2}$ s$^{-1}$, assumed boundary layer height of 1 km, and assumed accumulation time of one day.