# Peer review of "Wind-Driven Emissions of Coarse Mode Particles in an Urban"

_EGUsphere, 2023_

## Author Comment (AC1)

**Response to Referees**

Author statement: The authors thank the referees for their time reviewing this manuscript. An itemized response can be found below with each response given in blue. The tracked-changes version can be found below for each response in red.

**RC1: 'Comment on amt-2022-185', Anonymous Referee #1, 16 Jul 2022**

General comments:

The manuscript deals with the vertical flux of accumulation-mode and coarse-mode particles from the surface into the atmosphere. The authors use coherent Doppler lidar signals (at 1548 nm wavelength) backscattered from heights around 105 m above the ground to retrieve particle backscatter coefficients and vertical wind velocity (heterodyne technique). In addition, particle number concentrations are measured at ground level with an OPC. Thus, the reviewer recommends publication after addressing the below concerns.

Specific comments:

1. p2, line 31: How did you obtain the backscatter coefficient from the HALO Doppler lidar observations? I checked the paper of Chouza et al., Chouza, O. Reitebuch, S. Groß, S. Rahm, V. Freudenthaler, C. Toledano, and B. Weinzierl, Retrieval of aerosol backscatter and extinction from airborne coherent Doppler wind lidar measurements, Atmos. Meas. Tech., 8, 2909–2926, https://doi.org/10.5194/amt-8-2909-2015, 2015, and realized that this retrieval is a rather difficult approach. One has to consider turbulence effects and that the heterodyne efficiency is obviously range-dependent. And such a description is missing in your manuscript. Please provide a proper description of how you got the particle backscatter coefficient at 1.5 µm!

**Response:** Attenuated backscatter data were obtained directly from the DOE-ARM provided datafiles. According to their description (Newsom and Krishnamurthy, 2020), measurements are derived from the raw data using a factory-determined calibration curve. There is no further information provided about the procedure to derive the calibration or the uncertainties associated with the calibration. We now state this in the manuscript and provide additional checks further below.

**Manuscript (please see description further below):** Furthermore, the Doppler lidar backscatter is obtained via factory calibration (Newsom and Krishnamurthy, 2020). A back-of-the-envelope comparison of aerosol optical depth inferred from lidar backscatter against aerosol optical depth from the AErosol RObotic NETwork (AERONET) network at a nearby site shows that the lidar backscatter correlates strongly with aerosol optical depth, but its value may be biased high.

2. p3, line 1: D is what… ? Please write: …diameter D > 0.53 μm. I would use D_low instead of D_lo.

**Response:** Corrected

**Manuscript:** aerosol number fluxes for particles with diameter D > 0.53 μm are retrieved.

3. p4, section 2.4.2.: Why do you present this lengthy and quite complicated section on Mie scattering, when you, at the end, use the empirical correlations between attenuated backscatter from lidar at 105 m and particle number concentration measured in situ at ground?
Please reduce this part as much as possible! There are so many sources of uncertainty when applying Lorenz-Mie theory to large particles: a) large particles are usually non-spherical (they are irregularly shaped). Large particles are typical desert and soil dust particles and thus almost hydrophobic. What about the shape effect on the lidar ratio? If you take, for example, a Mie code to compute lidar ratios for spherical coarse-mode dust particles, you will end up with lidar ratios of 15-20 sr. But the real-world dust lidar ratio (for irregularly-shaped dust particles) at the wavelengths of 1064 nm and probably also at 1548 nm is 60-70 sr (Haarig et al., 2022). Mattis et al. (GRL, 10.1029/2002GL014721, 2002) showed already 20 years ago that the lidar ratio at 532 nm for non-spherical dust particles is around 50 sr and not around 20 sr. All this means: if you use Mie-modeled backscatter values these values may be wrong by a factor of 3 and thus the derived vertical aerosol fluxes may be wrong by a factor of 3.

**Response:** The referee's points are all correct, and we agree emphatically that the presented Mie model is insufficient to relate number concentration to aerosol backscatter. The referee further asks "[w]hy do you present this lengthy and quite complicated section on Mie scattering?". We present this section because it is *a-priori* surprising that aerosol backscatter should relate to aerosol number concentration. The Mie model provides justification that aerosol backscatter and aerosol number concentration are correlated (1) if an appropriate size-cut is used and (2) if the data are stratified by relative humidity. Thus the section is needed to motivate and support the application of empirical correlations to relate number concentration and backscatter.

**Manuscript: None**

4. p17, line 26 to p18, line 14: I would leave out this paragraph on INPs. This is just a jump into a completely different story. And this discussion makes only sense if the vertical flux of aerosol particles is directly connected with cloud evolution at the top of a well-mixed boundary layer (with cloud top temperatures of -10°C, -20°C or even -30°C…. and all this in June-August). By the way, $n_{aer,0.5}$ is not explained in this paragraph.

**Response:** We removed this section.

5. Figure 3: What results do you get for almost hydrophobic dust particles and with the true dust lidar ratio of 60-70sr at 1548 nm?

**Response:** We evaluated the model for $m = 1.6 + 0.06i$ and no water update, which produced a lidar ratio of 66. The result is shown below and is similar to the one shown in the manuscript. The correlations (not the values) are insensitive to input assumptions.

[Figure]

**Figure:** Statistical analysis of one day of aerosol size distributions. (a) Average aerosol size distribution measured by the OPC. The grey shading indicates the interquartile range. The red line shows a lognormal fit to the coarse mode. (b) Average size-resolved modelled $\beta_{mie}$, expressed as spectral density. The grey shading indicates the interquartile range. (c) Pearson correlation coefficient between $N(D_{dry} > D_{lo})$, the OPC integrated number of particles with diameter exceeding a lower threshold $D_{lo}$, and $\beta_{mie}$ at RH = 0%. (d) Correlation between integrated $\beta_{mie}$ (all sizes) and measured number concentration > 3 μm. Each point corresponds to a 30 min time average. Solid lines indicate a linear fit. Colours indicate the assumed relative humidity. (e) Same as panel (d) but for number concentration > 1 μm. (f) Same as panel (d) but for number concentration > 0.53 μm.

**Manuscript: The following change was made to the text**
Although the details change due to day-to-day variability of the shape of the size distribution, the assumed refractive index, and the assumed hygroscopicity, the overall trends in Figures 3(b)-(f) are repeatable.

6. Figure 4: Again the question arises: How did you obtain the backscatter values from the heterodyne Doppler lidar signals?

**Response:** Please see our response above.

7. Figure 5: So, obviously you get even negative backscatter coefficients! Please comment on that!

**Response:** We do not have negative backscatter and apologize for the confusing presentation. The graph shows fluctuations in vertical velocity and backscatter after applying the detrending (which also subtracts the mean). This was stated in the original text: *"Figure 5 shows an example of a contiguous block showing **detrended and despiked** vertical velocity and backscatter data.".* The intent was to show a block from which $\langle w'\beta'\rangle$ is computed. Unfortunately, the "prime" indicator to alert the reader, i.e. $w'$ and $\beta'$, were erroneously omitted in the figure, and appropriate labeling was missing in the caption. This is now corrected

**Manuscript: The following changes were made to the manuscript. (1) Updated figure, (2) updated caption.**

[Figure]

**Figure 5:** Example 780 s contiguous block of detrended and despiked vertical velocity ($w'$) and (b) backscatter ($\beta'$) data.

8. Figure 8: red values indicate backscatter coefficients of 30 to 50 Mm-1 sr-1. If we multiply these values with a 'real world' dust lidar ratio, we end up with aerosol extinction coefficients of 1800 to 3000 Mm-1. These are extinction coefficients of liquid-water clouds. Thick aerosol layers may show extinction coefficients up to 500 Mm-1 at 532 nm and up to 200 Mm-1 at 1548 nm.

What went wrong here?

The backscatter values should be < 10 Mm-1 sr-1 at 1548 nm wavelength, if not < 1 Mm-1 sr-1 during your observations, to my opinion.

**Response:** The reviewer is correct. We did point out in the text that the high backscatter values in the Figure likely correspond to low-lying (and in Houston summer at the base liquid) clouds). It is one of the main reasons that we do not consider vertical flux profiles in this work. We now superimpose the cloud-base height from the nearby ceilometer in the Figure, which confirms that the values in question are backscatter from clouds.

**Manuscript:** Updated Figure 8 and caption.

[Figure]

**Figure 8.** Three-day time series starting June 13 2022. (a) Time-height noise-thresholded attenuated backscatter curtain. Colours correspond to the log10 of the backscatter value in units of Mm-1 sr-1. The black solid line shows the cloud-based height retrieved by the ceilometer. Periods with no cloud base height data correspond to clear sky conditions. (b) Black: Backscatter flux at z = 105 m for each contiguous segment with z/L < 0 (unstable conditions), Grey: backscatter flux at z = 105 m for each contiguous segment with z/L < 0 (stable conditions). Red: limit of detection (LOD) computed using the lag method. (c) Stationarity metric ξ. The grey shading denotes the ±30% threshold given by Foken and Wichura (1996). (d) Hourly averaged fluxes for unstable conditions after the application of the flux loss correction. The grey shading indicates the combined uncertainty derived from $\sigma_{noise}$ + $\sigma_{sample}$ + $\sigma_{ensemble}$.

9. Figure 9: Again, time series of backscatter at 105 m height are shown. Please check AERONET 1640 nm AOD observations (Houston University) for June and July 2022 and compare these AERONET AODs with respective lidar-derived AODs when multiplying the backscatter coefficient (in km-1 sr-1) with 60 sr and with a boundary layer height of 1.5 km. Are the lidar-derived AODs close to the observed AERONET AODs for specific days? Yes or no… please mention this effort, i.e., the comparison with AERONET data, in the manuscript.

**Response:** This is an excellent suggestion. The Figure below shows the AOD from AERONET compared to that from LIDAR for the period. Note that AERONET coverage is limited to cloud-free conditions, while the LIDAR estimate can be made even under cloudy conditions. The two generally correlate well, although it is admitted that the boundary layer height and lidar ratio are less than suggested by the referee, and probably smaller than what is defensible based on general climatology (summertime convective boundary layer and dust aerosol). This comparison suggests that the calibrated DL backscatter may be biased low. However, the bias appears to be constant. Thus the calibration between DL-attenuated backscatter and aerosol number concentration via direct correlation corrects for the bias in DL-attenuated backscatter. This is now further discussed in the manuscript.

**Manuscript: The following was inserted into the manuscript:**

Furthermore, the Doppler lidar backscatter is obtained via factory calibration (Newsom. and Krishnamurthy, 2020). A back-of-the-envelope comparison of aerosol optical depth inferred from lidar backscatter against aerosol optical depth from the AErosol RObotic NETwork (AERONET) network at a nearby site shows that the lidar backscatter correlates strongly with aerosol optical depth, but its value may be biased high. Combined, these uncertainties are too large to rely on an optical model to relate measured backscatter to aerosol number concentration. Instead, this work relies on …

**The following was inserted into the supporting information:**

Comparison of aerosol optical depth (AOD) from the AErosol RObotic NETwork (AERONET) network at a nearby site and AOD derived from lidar. Here AOD from lidar was estimated from attenuated backscatter at $z = 105m$ ($\beta_{105}$) an assumed boundary layer height ($z = 1000m$) and a lidar ratio LR = 30 sr, using the approximation AOD = $\beta_{105}*z*$LR, which assumes that $\beta_{105}$ is representative of the entire boundary layer and that the lidar ratio is constant with altitude. Figure S2 shows the comparison between lidar-derived and directly observed AOD values. Note that AOD from AERONET is only reported for clear days, while AOD estimates from lidar are obtained for all days. Both boundary layer height and LR = 30 are likely lower than the actual values, suggesting that the absolute value of the measured backscatter may be biased high.

**Figure S2**. Comparison of aerosol optical depth from AERONET and aerosol optical depth derived from lidar.

10. Figure 10: F, Fflc, FwS, Fdep should be explained in the figure or in the figure caption.

**Response:** Done.

**Manuscript: The caption has been updated as follows.**

**Figure 10:** Temporal trend of daily averaged lidar retrieved daytime emission flux for particles D > 0.53 μm. Colours correspond to the contributions to the emission flux as given in Eq. (19). Here $F$ is the first order conversion from backscatter to number flux, $F_{flc}$ is the additional flux computed from the flux loss correction due to low frequency response of the Doppler lidar, $F_{wS}$ is the apparent contribution to the flux due to variation in the saturation ratio in updrafts and downdrafts, and $F_{dep}$ is the maximum estimated contribution of the deposition velocity to the flux. White areas correspond to dates where fluxes were below the detection limit or insufficient data were available to compute flux corrections.

11. Figure 11: We need this figure for the most trustworthy backscatter values….

**Response:** We only included backscatter **flux** values that are trustworthy based on our uncertainty analysis.

**Manuscript: None.**

12. Figure 12: I would remove this figure plus the discussion, as mentioned already. This is not needed in this paper. You may consider it in a follow up paper on aerosol-cloud interaction.
The INP dependence on temperature is confusing. Usually, INP efficiency increases by an order of magnitude when temperature decreases by 5 K, at least for temperatures from -20°C to -35°C. This is by far not the case in Figure 12.

**Response:** Removed as requested.

**RC2: 'Comment on egusphere-2023-951', Anonymous Referee #2, 30 Oct 2023**

General comments:

This paper describes the use of surface-based Doppler lidar measurements to retrieve the surface number particulate emissions for aerosol particles larger than 0.53 micrometers in diameter. The technique is applied to Doppler lidar measurements acquired near Houston, TX. The paper describes this method and how it used surface-based optical particle counter measurements of aerosol size distribution to calibrate the Doppler lidar measurements of near-surface aerosol backscatter. The paper presents a new method for retrieving coarse mode surface number emissions. The paper is well written, and the figures are quite adequate for presenting the results and for illustrating the discussion. I recommend publication after the authors address the relatively minor questions and comments given below.

Specific comments:

1. Page 1, Line 12. It would be helpful to indicate when the two-month period occurred.

**Response:** Corrected

**Manuscript:** "..over a two-month period from June 1 to August 10, 2022 during.."

2. Page 4, equation 1, I believe this expression assumes the lidar ratio does not vary with range; it would be helpful to indicate this in the text.

**Response:** Correct.

**Manuscript:** … LR is the lidar ratio defined in Eq. (4), and the integration is carried out between ranges r1 and r2. In Eq. (1) it is assumed that the lidar ratio does not vary with range.

3. Page 4, line 18. Here S represents lidar ratio. Later, in equation 15, S represents saturation ratio. It would be helpful if an alternative method to represent lidar ratio and/or saturation ratio was used to avoid confusion.

**Response:** We changed S to LR to differentiate the lidar ratio.

**Manuscript: Changed Eq. (1), text, and Figure 2.**

4. Page 4, line 2. What absorbing gases are present at the laser wavelength? I would assume absorbing gases would typically not be a factor.

**Response:** The reviewer is correct that there are few absorbing gasses that are likely not an issue. None of the major gasses (e.g. $O_2$, $CO_2$, $O_3$, $CH_4$) absorb at the DL wavelength (c.f. Figure 1 in Patadia et al., 2018). However, we do not know if some trace gasses may absorb at this wavelength and thus we have this possibility enumerated.

Patadia, F., Levy, R. C., and Mattoo, S.: Correcting for trace gas absorption when retrieving aerosol optical depth from satellite observations of reflected shortwave radiation, Atmos. Meas. Tech., 11, 3205–3219, https://doi.org/10.5194/amt-11-3205-2018, 2018.

5. Page 5, line 6. The authors note that the Mie solution for the relationships shown on this page assumes that the particles are spherical. However, given that coarse mode dust particles are often nonspherical, can the authors comment on the applicability of this analysis, especially since they later admit that the uncertainties associated with the optical model are too large to relate observed aerosol backscatter to particle number concentration? Also, given that these uncertainties are too large, and the later analyses rely on empirical correlations with surface OPC measurements, what is the point of this Mie analysis?

**Response:** We present this section because it is *a-priori* surprising that aerosol backscatter should relate to aerosol number concentration. The Mie model provides justification that aerosol backscatter and aerosol number concentration are correlated (1) if an appropriate size-cut is used and (2) if the data are stratified by relative humidity. Thus the section is needed to motivate and support the application of empirical correlations to relate number concentration and backscatter.

**Manuscript: None.**

6. Page 5, line 6. Regarding the particle nonsphericity, the ARM measurements also included measurements of aerosol backscatter and depolarization by a Micropulse Lidar (MPL). Could these measurements of depolarization be used to provide some indication of the prevalence of nonspherical particles?

**Response:** Yes, this may be a possibility and we are working on methods that use/assimilate information from a second LIDAR system. A potential complication is that the MPL operates at a different wavelength (532 nm), and thus is sensitive to a different slice of the aerosol size distribution.

**Manuscript: None.**

7. Page 5. This analysis also relies on the assumption of the refractive index of the particles, which depends on particle composition. Typically ARM also measures particle composition at the surface in these ARM AMF deployments. Were there no measurements of coarse mode particle composition available?

**Response:** We are not aware of supermicron composition analysis available for those periods.

**Manuscript: None.**

8. Page 7, line 11. The analyses that depend on RH use RH determined from interpolation from radiosondes. How often were the radiosondes launched?

**Response:** The following was added to the manuscript.

**Manuscript:** For upper air observation data, ARM provided interpolated sonde data containing relative humidity, specific humidity, temperature, horizontal wind, potential temperature, and dew point temperature on a fixed time-height grid. The data has 332 levels with a 1-minute time resolution from the surface to a maximum of about 40 km. It is based on 4 launches per day.

9. Page 8, line 24. I think the sentence should say that the spectrum for backscatter is flat for frequencies above 0.035 Hz; the spectrum for vertical velocity does not look flat for frequencies above 0.05 Hz so I don't follow the discussion at the top of page 9.

**Response:** The text was revised as follows.

**Manuscript:** The spectra for $S(\beta)$ are flat for frequencies larger than 0.035 Hz, which indicates white noise. Integrating $\int S(\beta)df$ from 0.035 Hz to the Nyquist frequency equals the noise variance $\delta^2_\beta$ derived from the autocovariance analysis. Conversely, the spectra for $S(w)$ do not flatten. Thus the transition to white noise cannot be used to determine the noise limit. An estimate of the noise variance $\delta^2_w$ can be found by integrating $\int S(w)df$ from 0.05 Hz to the Nyquist frequency, which equals the noise variance $\delta^2_w$. The visual depiction and the magnitude of these thresholds are similar for other contiguous segments.

10. Page 11, line 18. If interpolated radiosondes were used to provide the RH at z=105 m, why weren't these interpolated radiosondes also used to provide the temperature at z=105 m instead of using the surface temperature measurements?

**Response:** We used the surface temperature data since it has better time resolution and since temperature changes over 100 m are less pronounced than humidity changes. However, the calculated footprints are not sensitive to this choice.

**Manuscript:** The surface temperature from the surface meteorological station at the site was used as the temperature at $z = 105$ m due to the lack of high-resolution temperature data at that height. However, the calculated flux footprints are not sensitive to the choice of temperature variable.

11. Page 14, line 28. Figure 9 shows some backscatter values exceed 10 (Mm-sr)-1. Assuming lidar ratios around 50 sr leads to extinction values around 500 Mm-1 which are very large; too large for soil dust or biological activity. Were there local sources of dust nearby? The DOE ARM TRACER campaign field report (https://www.arm.gov/publications/programdocs/doe-sc-arm-23-038.pdf) states that several ARM and guest instruments and NASA GMAO models indicated that Saharan dust was observed one or more occasions. The report indicates that one of the events occurred on 17-18 July which coincides with the peak in aerosol backscatter shown in Figure 9 and so can possibly explain such large values. Can the authors please comment on the presence and impact of such aerosols on the derived number concentrations and fluxes?

**Response:** We already briefly commented on this in the text, without attributing these to explicit dust events: "Nevertheless, the retrieval captures the broad trends in particle number concentration, including the transition from lower concentration to higher concentration periods 11 June, 15 July, and 20 July." We avoided attribution to Saharan dust since we do not have any composition data. For the purpose of this paper, we were satisfied that the natural fluctuations in number concentration, including the passage of one or more "dust events", were captured by the retrieval. We were also satisfied that there was no obvious correlation between aerosol number concentration and aerosol number flux, suggesting that horizontal advection can be separated from surface emissions.

12. Page 17. The last part of the discussion deals with role of coarse mode particles on INP. Page 18,lines 5-6 mention how the estimates of INP impacted by such near surface coarse mode particles are sensitive to many factors. Given such (large) uncertainties and the lack of validating INP data, I suggest that this discussion of INP be omitted.

**Response:** The section was removed as suggested by the reviewer.